# Structural basis for the toxicity of *Legionella pneumophila* effector SidH

Rahul Sharma [1], Michael Adams[1], Simonne Griffith-Jones [1], Tobias Sahr[2], Laura Gomez-Valero[2], Felix Weis [3], Michael Hons [1], Sarah Gharbi [1], Rayene Berkane [4,5], Alexandra Stolz [4,5], Carmen Buchrieser [2] & Sagar Bhogaraju [1] ✉

*Legionella pneumophila* (*LP*) secretes more than 300 effectors into the host cytosol to facilitate intracellular replication. One of these effectors, SidH, 253 kDa in size with no sequence similarity to proteins of known function is toxic when overexpressed in host cells. SidH is regulated by the *LP* metaeffector LubX which targets SidH for degradation in a temporal manner during *LP* infection. The mechanism underlying the toxicity of SidH and its role in *LP* infection are unknown. Here, we determined the cryo-EM structure of SidH at 2.7 Å revealing a unique alpha helical arrangement with no overall similarity to known protein structures. Surprisingly, purified SidH came bound to a *E. coli* EF-Tu/t-RNA/GTP ternary complex which could be modeled into the cryo-EM density. Mutation of residues disrupting the SidH-tRNA interface and SidH-EF-Tu interface abolish the toxicity of overexpressed SidH in human cells, a phenotype confirmed in infection of *Acanthamoeba castellani*. We also present the cryo-EM structure of SidH in complex with a U-box domain containing ubiquitin ligase LubX delineating the mechanism of regulation of SidH. Our data provide the basis for the toxicity of SidH and into its regulation by the metaeffector LubX.

*Legionella pneumophila* (*LP*) is a gram-negative bacterium that is pathogenic for humans when it reaches the respiratory tract where it infects lung macrophages to cause an acute pneumonia called Legionnaires' disease[1]. However, *LP* is an environmental bacterium and various species of protozoa, mainly amoeba are its natural hosts in which they replicate intracellularly. Interestingly, *LP* has the largest repertoire of effector proteins known among pathogenic bacteria. *LP* uses its Type IV Dot/Icm secretion system (T4SS) to release more than 300 effector proteins into the host cytosol during infection, which can modulate diverse host cell pathways and successfully establish a replicative niche inside the *LP* containing vacuole (LCV)[2]. A vast number of *LP* effectors have been shown to hijack the host vesicular transport machinery to recruit ER derived vesicles that fuse with the growing LCV[3]. In addition to secretion of proteins during infection, *LP* has also recently been found to release extra cellular vesicles containing bacterial tRNAs and long non-coding RNAs that directly modulate the host immune response[4].

The large arsenal of *LP* effectors contains protein families that redundantly and additively work on similar host defense pathways[5–7]. Therefore, for the majority of the effector proteins, deletion of any single effector protein from the *LP* genome does not dent the bacteria's replicative capacity inside the host. In contrast, one *LP* effector protein called SdhA was shown to be essential for the replication of *LP* inside mouse macrophages[8]. SdhA belongs to the SdhA family of

[1]European Molecular Biology Laboratory, 71 avenue des Martyrs, 38042 Grenoble, France. [2]Institut Pasteur, Université Paris Cité, Biologie des Bactéries Intracellulaires and CNRS UMR 6047, 75724 Paris, France. [3]European Molecular Biology Laboratory, Meyerhofstraße 1, 69117 Heidelberg, Germany. [4]Institute of Biochemistry II, Goethe University Frankfurt - Medical Faculty, University Hospital, Frankfurt am Main, Germany. [5]Buchmann Institute for Molecular Life Sciences, Goethe University Frankfurt, Frankfurt am Main, Germany. ✉e-mail: bhogaraju@embl.fr

effectors sharing ~40% sequence similarity with its paralogs SdhB and SidH in the N-terminal region (~700 amino acids) of the protein sequence[9]. Interestingly, deletion of SdhA alone results in defective growth in mouse macrophages while deletion of the two paralogues SidH and SdhB does not cause any significant growth phenotype indicating that these paralogs may have different functions during infection. Further studies showed that deletion of SdhA leads to host cell apoptosis upon infection[8] and that SdhA is important for the integrity of the LCV. It was also suggested that the lack of LCV integrity may lead to the release of bacterial nucleic acids into the host cytosol triggering the foreign DNA/RNA sensing innate immunity machineries in the host[10–14]. A recent report by Choi et al. showed that SdhA interacts with OculoCerebroRenal syndrome of Lowe (OCRL), a lipid phosphatase that acts on phosphatidylinositol-4,5-bisphosphate $(PtdIns(4,5)P_2)$ to remove the 5-position phosphate[12]. OCRL plays an important role in endosomal trafficking and crucially also in the autophagosome-lysosome fusion[15]. It was found that deletion of OCRL from the host cells partly rescues the LCV integrity phenotype found during the infection with the $LP \Delta sdhA$ mutant strain[12]. More recently, it was also found that while SdeA family of $LP$ effectors guard LCV in the first few hours of infection, SdhA plays an essential role of guarding LCV at later stages of infection[16].

The function of the SdhA paralogs SidH and SdhB remains less explored. SidH is 253 kDa protein that has been shown to be toxic when ectopically overexpressed in yeast in large scale toxicity screening studies[17]. It was later found that SidH is regulated by another effector protein of $LP$ via the host ubiquitin system during infection[18]. Ubiquitination of proteins is a conserved post translational modification which involves the covalent attachment of a conserved 76 amino acid polypeptide, called ubiquitin, to target lysine residues of the substrates through an isopeptide bond. The reaction of protein ubiquitination proceeds through a cascade of three enzymes named Ubiquitin-activating (E1), Ubiquitin-conjugating (E2), and Ubiquitin ligase (E3) which finally facilitates the transfer of activated ubiquitin to the substrate lysines[19]. Bacteria do not have their own ubiquitin system but many intracellular bacteria possess effector proteins that intercept many of the host signaling events by acting as ubiquitin ligases, deubiquitinases (DUBs) or ubiquitin modifying enzymes[20]. $LP$ contains many canonical and one non-canonical system of ubiquitin ligases that hijack the host ubiquitin system[21]. One of the $LP$ effectors known as LubX contains two Ubox domains which are the catalytic domains present in a class of ubiquitin ligases found in eukaryotes. LubX uses one of its Ubox domains to ubiquitinate SidH in a temporal manner causing its degradation during $LP$ infection. Furthermore, LubX is the first so called "metaeffectors" identified, an effectors that regulates the activity of other effectors in a timely manner during $LP$ infection[17]. Now it has been shown that $LP$ harbors several such metaeffectors[18,22,23]. Interestingly, deletion of SidH or LubX does not lead to any intracellular growth phenotype during $LP$ infection of primary mouse macrophages[8,24]. However, Kubori et al. performed Drosophila infection experiments with various deletion strains of $LP$ and this uncovered that the flies died more readily when infected with the $\Delta lubX$ strain compared to the WT and $\Delta sidH$ strains. This is consistent with the idea that SidH is toxic for host cells and persistent SidH toxicity during infection with the $\Delta lubX$ strain may be unfavorable for $LP$ replication[18]. The reasons of the toxicity of SidH and the basis for the selective ubiquitination of SidH by LubX are unknown.

Because SidH and the paralogs share no notable sequence similarity with proteins of known function, biochemical characterization of these proteins remained difficult. Here we determined the cryo-EM structure of SidH at a nominal resolution of 2.7 Å. SidH folds into a large network of helical bundles that does not share a notable similarity to any other protein in the protein data bank (PDB). Interestingly, we found that SidH interacts with bacterial tRNA and EF-Tu as it comes bound to these molecules during the purification. We found that

ectopic overexpression of SidH is toxic to human cells and the toxicity is completely lost when the interface of SidH-tRNA or SidH-EF-Tu is disrupted, a phenotype that could be confirmed during infection of $A.$ $castellani$ with these different mutants. SidH expressed in HEK293T cells co-immunoprecipitated with ribosomal proteins and other tRNA-associated and regulatory proteins, however SidH does not seem to affect protein synthesis in vitro or global protein translation in cells. Finally, we also determined a high-resolution cryo-EM structure of SidH in complex with LubX and show how the C-terminal U-box2 domain of LubX interacts with SidH while the U-box1 domain remains flexible in the SidH-LubX complex to execute ubiquitination.

## Results

### Cryo-EM structure of SidH

In order to gain insights into the biochemical and physiological function of SidH (encoded by $lpg2829$ in strain Philadelphia), we sought to determine its structure using Cryo-Electron microscopy (Cryo-EM). Recombinant SidH with an N-terminal His tag was expressed in $Escherichia\ coli\ BL21(DE3)$ and purified as described in the methods section (Fig. S1A). Interestingly, we found that an $E.\ coli$ protein of 45 kDa co-purified with SidH. Mass photometry analysis showed that the purified SidH was homogenously containing only one major population but the estimated molecular weight was ~324 kDa which is 70 kDa higher than the theoretical molecular mass of SidH (Fig. S1B). To identify the unknown protein that co-purified with SidH contributing to the apparent mass gain of SidH, we employed peptide mass fingerprinting (Fig. S1C). This revealed that the $E.\ coli$ elongation factor thermo unstable (EF-Tu) (Mw of 45 kDa), a protein catalysing the loading of aminoacyl tRNA onto the ribosome, interacts with SidH and co-elutes during the purification procedure (Fig. S1C). The basis for the remaining unexplained mass gain (~25 kDa) of purified SidH was not revealed at this point but the structure helped us gain insights into this (see below).

The purified SidH was analyzed using single particle cryo-EM yielding a density map with a nominal resolution of 2.7 Å (Fig. 1a, S2). The cryo-EM map (Fig. 1a) revealed that SidH is an α-helical protein. Interestingly, a section of the map resembled an "L" shaped double helical density typical of a tRNA molecule (Fig. S3). We could indeed fit a tRNA model into this "L" shaped density indicating that both EF-Tu and tRNA came bound to the purified SidH. Accordingly, we could morph the model of tRNA$^{phe}$-EF-Tu complex (PDB ID- 1TTT) into a part of the obtained cryo-EM density (Fig. S4).

As SidH does not have any homology to proteins of known structure, we built the model de novo using a combination of Phenix autobuild package and manual building in Coot. A small region spanning residues 903-1040 of SidH was modeled using the AI structure prediction tool AlphaFold[25]. The final refined structure contains one molecule each of $LP$ SidH and $E.\ coli$ EF-Tu/tRNA complex (Fig. 1b). The structure of SidH is composed of a set of discrete bundles of helices that are intricately arranged with respect to each other, resulting in the overall shape of SidH resembling the letter "I". For the sake of describing the structure of SidH and its interaction with t-RNA and EF-Tu, we defined 8 distinct helical bundles (Hb) in SidH (Fig. 1c). These are Hb1 (1-175), Hb2 (189-339), Hb3 (382-546), Hb4 (548-709), Hb5 (710-874), Hb6 (903-1040), Hb7 (1241-1369) and Hb8 (1388-1619) (Fig. 1c). The N-terminal region of SidH containing Hb1-4 forms an extended helical domain which lies roughly parallel to a similar helical domain formed by the C-terminal region containing Hb7-8 of the protein. Interestingly, the N- and the C-terminal regions of SidH share 33% sequence homology hinting at a gene duplication event during the evolution of SidH (Fig. S5). The N- and C-terminal parallel helical domains are joined by the intermediate Hb5. Hb6 of SidH uniquely points away from the core of SidH and is likely flexible with respect to the rest of the protein as evident from the relatively poor local resolution in this region (Fig. S6A). We used local classification and

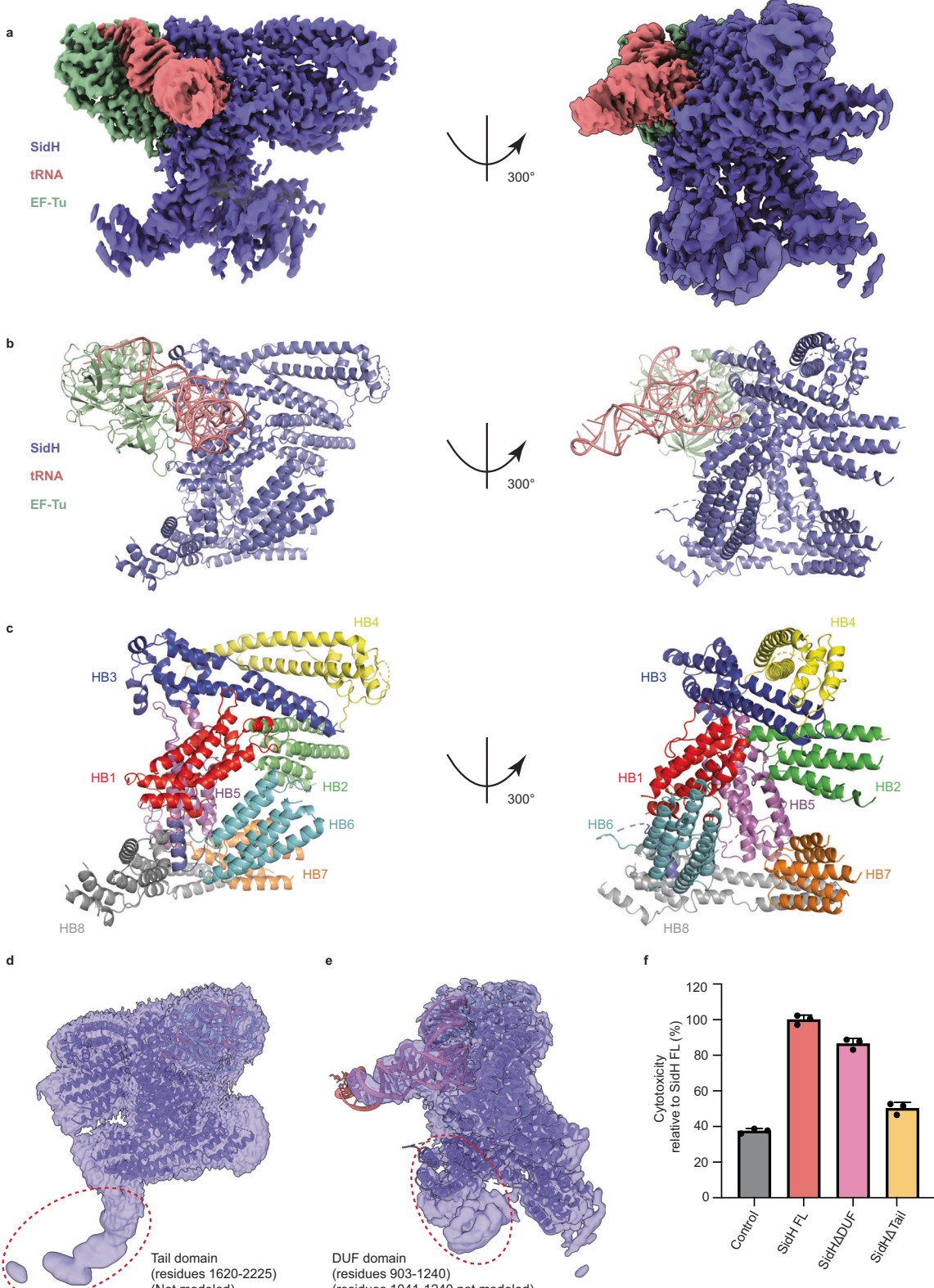

**Fig. 1 | Overall structure of SidH in complex with tRNA^Phe and EF-Tu. a** A cryo-EM map of the complex showing densities for SidH (blue), tRNA^Phe (pink), and EF-Tu (green). **b** A complete model of the complex showing cartoon representation of SidH (blue), tRNA^Phe (pink), and EF-Tu (green). The residues not modeled in the structure are shown by dashed lines. **c** The model of SidH is colored according to the distinct helical bundles (Hb1-Hb8). **d** Unsharpened EM map shows poor resolution in the region of Tail domain (circled with red dotted line). The SidH residues (1620-2225) could not be modeled in this region. **e** Unsharpened EM map shows poor resolution in the region of DUF domain (circled with red dotted line), which includes residues from 903 to 1240 of SidH. Although an AlphaFold model for the residues 903-1040 could be fitted, an almost 2/3rd part (1041-1240) of this region could not be modeled. **f** Cytotoxicity assay shows toxicity levels of SidHΔDUF and SidHΔTail relative to the toxicity of SidH full-length (FL). Data represents the mean ± SD of three independent reactions. Source data are provided as a Source Data file.

refinement in Relion[26] to better resolve this region of SidH and used AlphaFold[25] to model Hb6 (Fig. S6B).

Two major regions of SidH located in the central (1041-1240) and the C-terminal (1620-2225) parts could not be built into the EM map due to the poor local resolution (5–8 Å) of the map in these regions (Fig. 1d, e, S6). As seen in the unsharpened cryo-EM map, the C-terminus of SidH spanning residues 1620 to 2225 adopts a tail shaped architecture (Fig. 1d) and forms a distinct domain in SidH. We named this C-terminal unmodeled region as the Tail domain of SidH. The central unmodeled region (residues 1041-1240) contains a small but distinct α-helical subdomain (Fig. 1e) that closely interacts with Hb6 to form a distinct domain of SidH. We named this domain containing Hb6 and the region spanning residues 903-1240, the domain of unknown function or DUF (Fig. 1e).

For structure-based prediction of SidH function, we compared the final SidH model to the protein structures in the PDB using the DALI server[27]. The DALI search did not yield any major structural homologues of SidH with a notable alignment and r.m.s.d. Interestingly, Hb6 of SidH shares structural homology with translocator proteins (TSPO) with an r.m.s.d of 3.2 Å over 118 C-α s (Fig. S7A). Both TSPOs and Hb6 of SidH contain 5 helices arranged cylindrically. TSPOs bind various benzodiazepine ligands including the medically relevant human TSPO which binds to cholesterol and plays a crucial role in steroidal synthesis in neuronal mitochondria[28]. However, the sequence homology between SidH Hb6 and the closest TSPO homologue from *Rhodobacter sphaeroides* is only 6%. Moreover, TSPOs are transmembrane proteins whereas SidH Hb6 does not have any characteristics of a transmembrane domain. Because of these factors, we predict that SidH Hb6 likely has a different function compared to TSPOs. A lack of significant structural homologues of SidH in the PDB indicates that the structure of SidH is unique and it might not be possible to deduce the function of SidH only through structural homology to proteins of know function.

## Contribution of DUF and the Tail domains towards the toxicity of SidH

Ectopic expression of many *LP* effectors in yeast is toxic and can be used as a readout to map the functionally important regions of toxic effectors. It has been shown previously that the expression of SidH leads to lethality in yeast cells[17]. We observed that SidH expression also leads to toxicity in human HEK293T cells which is manifested as rounding of cells and can be quantified using the ToxiLight assay kit (Lonza), which measures the release of Adenylate cyclase into the culture media from lysed cells (Fig. 1f, Fig. S7B). Deletion of the tail domain in SidH nearly abolished the toxicity of SidH indicating that this domain may be of functional importance for SidH. Deletion of the SidH DUF domain however did not affect the toxicity in HEK cells notably. This shows that the DUF domain of SidH does not contribute greatly to the toxicity of SidH, though we cannot however exclude a functional role for the SidH DUF domain during *LP* infection (Fig. 1f).

## SidH-tRNA interaction

Although the cryo-EM structure of SidH unambiguously shows the binding of tRNA to the N-terminal helical bundle of SidH, cryo-EM density alone was not sufficient to discern which specific tRNA is bound to SidH. *E. coli* contains more than 70 genes which encode for different tRNA isotypes. In order to identify the type of tRNA SidH binds to, we subjected the tRNA isolated from purified SidH to RNA sequencing analysis. These data showed that SidH can bind to multiple tRNAs and that binding to tRNA is sequence independent (Fig. 2a). Accordingly, the nucleotide sequence could not be deduced from the EM map in the variable regions of tRNA presumably because the final cryo-EM map of SidH involved averaging of many SidH particles bound to different tRNAs. Therefore, for the purpose of analyzing the structure, we placed phe-tRNA into the map as most of the nucleotides fit agreeably into the electron density.

tRNA and SidH share a buried surface area of 500 Å$^2$ which is mostly composed of ionic interactions. The T-loop situated at the kink of the "L" shaped tRNA molecule inserts itself into the positively charged groove formed at the intersection of Hb1, Hb3 and Hb5 in the N-terminal region of SidH (Fig. 2b, c). Interestingly, the T-loop is one of the most conserved features among tRNAs both in terms of sequence and structure explaining why SidH does not show specificity to one particular tRNA (Fig. S8). The acceptor arm of the tRNA does not make any direct contact with SidH and reaches into the β-barrel domain 2 of EF-Tu in a similar configuration as it does in the EF-Tu/Phe-tRNA/GDPNP ternary complex (PDB: 1TTT). The anticodon arm of the tRNA extends into the solvent away from SidH and has poorly resolved electron density (Fig. 2b, c, S6). With regards to specific interactions between SidH and tRNA, Arg819 of SidH interacts with the phosphates of U55, G57 and C56 of the tRNA (Fig. 2d). Lys110, Lys504 interact with the phosphates of G53 and G63 respectively. Lys71 and Lys57 of SidH stack on to the bases of C56 and G19 (situated in the D-loop) of tRNA respectively.

## SidH/EF-Tu interaction

EF-Tu that is bound to the tRNA in the structure of SidH has clearly resolved Guanosine triphosphate (GTP) in its active center (Fig. 3a) indicating this ternary complex (EF-Tu/tRNA/GTP) is in principle ready to translocate tRNA onto the A-site of the translating ribosomes. EF-Tu and SidH share a buried surface area of 1184 Å$^2$ where the N-terminal region of SidH mediates interaction with the GTP-binding Domain 1 and the beta-barrel Domain 3 of EF-Tu (Fig. 3b–d). There are 2 distinct contact sites between SidH and EF-Tu that extend along the length of the N-terminal half of SidH (Fig. 3b). Site 1 links the GTP-binding Domain 1 of EF-Tu to Hb3 of SidH (Fig. 3c). At site 1, W438 and K437 of SidH Hb3 point directly towards the GTP-binding pocket situated in the Domain 1 of EF-Tu. SidH W438 engages catalytic His85 of EF-Tu in a cation-pi interaction and also makes hydrophobic contacts with I61 and the aliphatic part of the R59 side chain. K437 of SidH is involved in a salt bridge interaction with D22 of EF-Tu. The GTP-binding domain of EF-Tu is additionally tethered to SidH via hydrophobic interactions between V486 of SidH and V141, L146 of EF-Tu. Also, K495 of SidH hydrogen bonds with the backbone carbonyl groups of D110 and P112 while the aliphatic part of K495 makes hydrophobic contacts with M113 of EF-Tu. Site 1 also involves a hydrogen bond between Q496 of SidH and the backbone carbonyl group of D315 from the beta-barrel domain 3 of EF-Tu.

At site 2 of the SidH/EF-Tu interaction, the beta-barrel Domain 3 of EF-Tu interacts with Hb1 of SidH mainly through hydrophobic contacts. F324, V348, M350, M352 of EF-Tu form a hydrophobic interface with M106, Y109, L151, A155 and P156 of SidH Hb1 (Fig. 3d).

A charge reversal mutation of positively charged residues lining the tRNA binding groove (SidH hexamutant (HM) K57E, K71E, K110E, A117E, K504E and R819E) resulted in loss of binding to both tRNA and EF-Tu molecules (Fig. 3e).

## Evolution of the sidH gene within the species of *Legionella pneumophila*

We noticed that the *sidH* gene in the *LP* strain Paris is split into two genes *lpp2886* and *lpp2883* coding for amino acids 1-1025 (SidH$^{Paris}$ N-term) and 1026-2225 (SidH$^{Paris}$ C-term), respectively, due to the presence of an insertion sequence (Fig. 4a). Moreover, we found that the *sidH* gene is absent in strains Lens[29] and Corby[30]. Intrigued by these observations, we undertook a comparative and evolutionary analyses of the *sidH* gene within the species *L. pneumophila*. We retrieved 973 *LP* genomes from the RefSeq database and searched for *sidH* in these strains using *sidH* Philadelphia (*lpg2829*) as reference and blastn. *sidH* was present and highly conserved (more than 95% nucleotide identity) in 775 strains, absent in 174 strains and shorter or fragmented in 24 strains. In the 24 strains with fragmented *sidH* the gene was

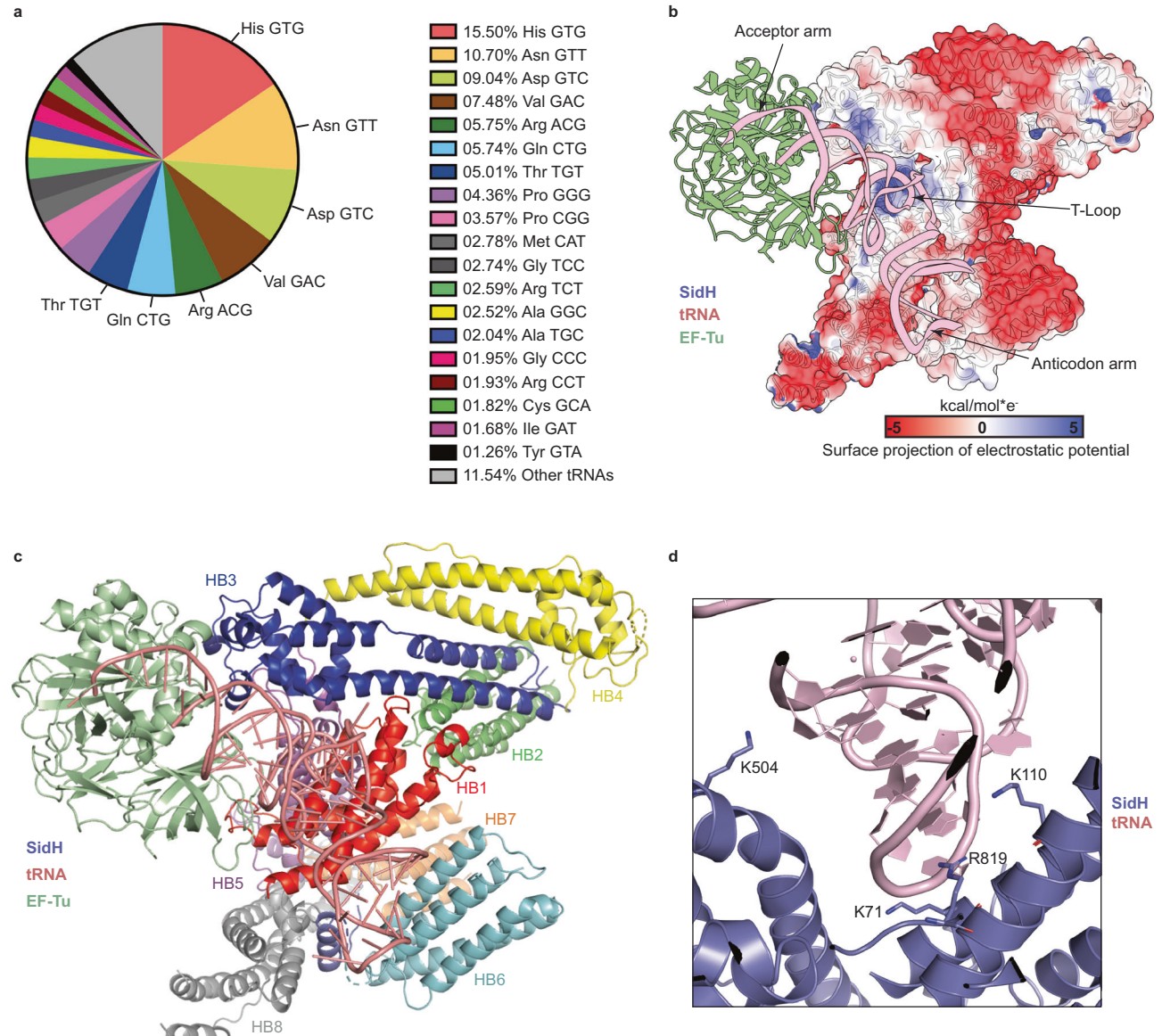

**Fig. 2 | SidH binding to tRNA is sequence independent. a** The pie chart shows different type of tRNAs came in bound state to purified SidH from *E. coli*. tRNA isotypes with their percentage of abundance and color scheme (right panel). Source data are provided as a Source Data file. **b** The electrostatic potential surface projection of SidH shows that T-loop of tRNA (pink) inserts into positively charged groove (blue) of SidH. **c** The cartoon representation of the complex shows that the binding groove for tRNA is composed of distinct helical bundles colored and labeled as explained in the main text. **d** The detailed view of tRNA-binding region of SidH shows that the side chains of positively charged residues are pointed towards negatively charged phosphodiester backbone of tRNA to make ionic interactions.

interrupted by transposases only in strains Paris, C3 and E8. In strains C3 and E8 the *sidH* fragments are identical and are interrupted by the same IS4-like element, whereas in strain Paris the size of both fragments is slightly different and it is interrupted by a different transposase (Fig. S9). The phylogenetic distribution shows that the C3 and E8 strains cluster together in the tree but are distant from strain Paris indicating two different events of transposase insertions: one in the ancestor of C3 and E8 and another in the branch leading to the Paris strain (Fig. S10). This observation suggests that *sidH* was probably present in the common ancestor of *L. pneumophila* strains and has then been lost independently many times during the evolution of this species.

SidH from the Paris and Philadelphia strains share very high sequence similarity (98%) and all residues involved in the binding of tRNA and EF-Tu are identical between these two homologues. To understand if the SidH[Paris] is still functional, we expressed and purified the two individual fragments of SidH[Paris] (Fig. S11A, S11C). As expected,

SidH[Paris] N-term which harbours the tRNA and EF-Tu binding region came bound to these factors during the purification procedure similar to SidH Philadelphia (Fig. S11A). Intriguingly, a notable amount of purified SidH[Paris] N-term was also devoid of tRNA and EF-Tu (Fig. S11B). This is perhaps due to the relatively high expression of SidH[Paris] N-term in *E. coli* compared to SidH full-length (FL) from the Philadelphia strain. We first checked whether SidH[Paris] N-term and SidH[Paris] C-term interact with each other in vitro. Indeed, these two fragments of SidH[Paris] eluted together when co-injected into an analytical size-exclusion column (Fig. 4b). We further checked the toxicity of SidH[Paris] fragments in HEK cells. SidH[Paris] N-term or SidH[Paris] C-term when overexpressed alone are not toxic to HEK cells but co-expression of these two fragments in HEK cells led to severe toxicity comparable to that of SidH Philadelphia (Fig. 4c, Fig. S12A). This indicates that tRNA/EF-Tu binding is essential but not sufficient for the toxicity of SidH; SidH also needs the presence of its C-terminal region to exert its full toxic effect in cells. It is possible that the C-terminus of SidH plays a structural role and that it is

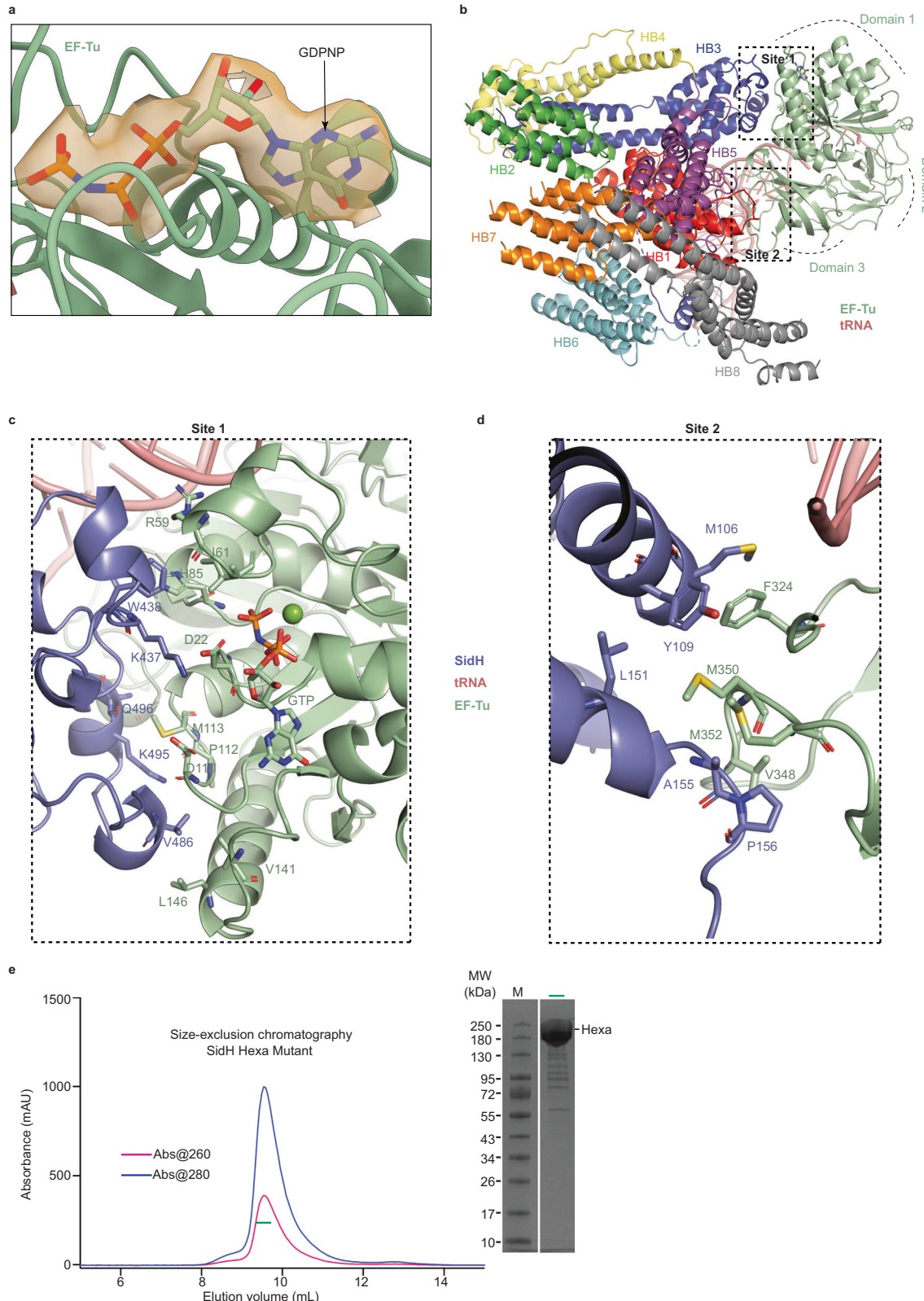

**Fig. 3 | Binding interface between SidH and EF-Tu. a** A molecule of GDPNP (stick representation) bound to EF-Tu (cartoon). The electron density for the GDPNP molecule is shown. **b** Extensive interaction interface between SidH and EF-Tu is shown in black dotted rectangles as site1 and site2. **c** Detailed view of interactions at site1- key residues (stick representation) which are involved in interaction between domain1 of EF-Tu (green) and Hb3 of SidH (purple) are shown. **d** Similar view as *panel C* highlighting the interactions at site2- interaction between domain 2 of EF-Tu (green) and Hb1 of SidH (blue). Stick representation of key residues are shown. **e** Size-exclusion chromatography (SEC) of Hexa mutant (a tRNA binding deficient mutant of SidH)- the ratio of absorbance (260/280 nm) shows the absence of tRNA. The labelled peak fraction (a horizontal green bar) was loaded on SDS-PAGE gel (right panel), which shows absence of EF-Tu as well, as there is no protein band at around 43 kDa of molecular weight marker. Source data are provided as a Source Data file.

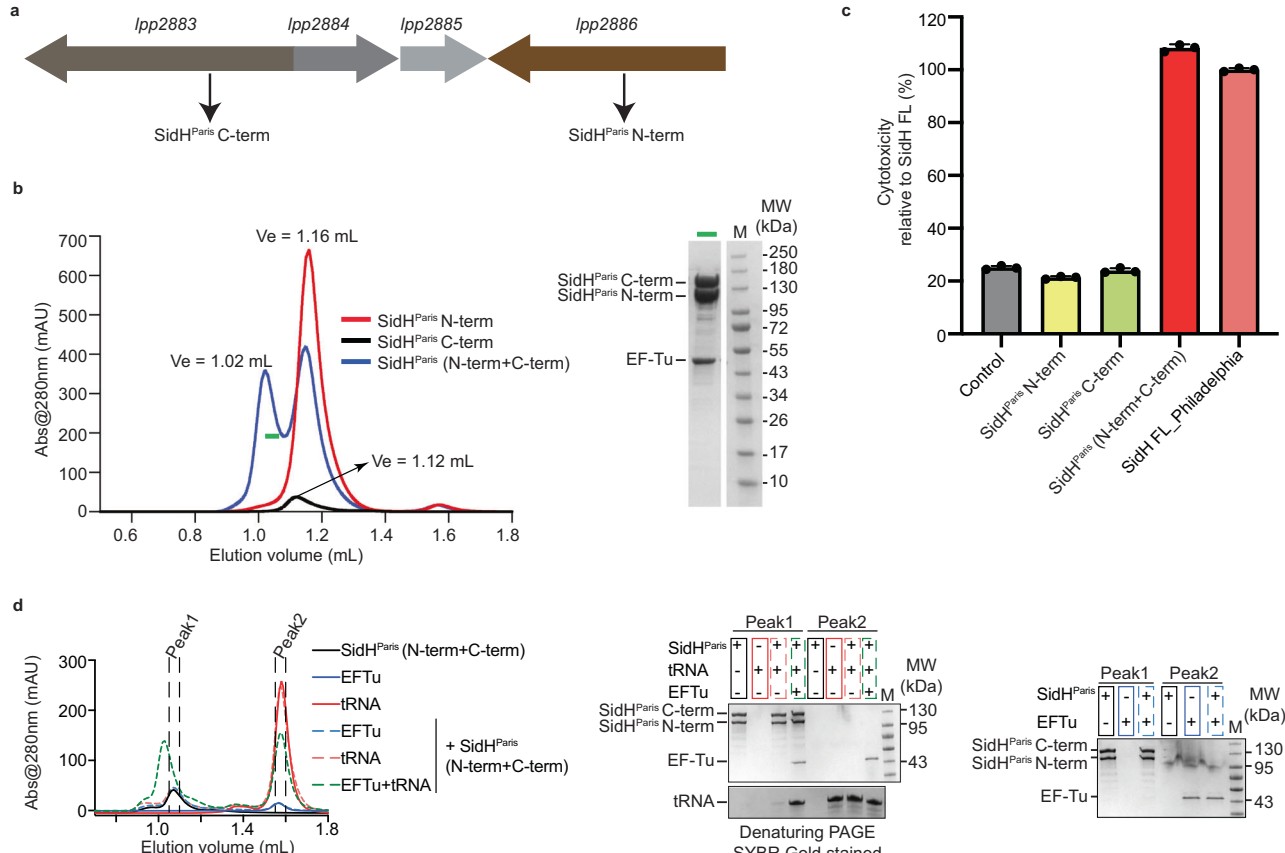

**Fig. 4 | Split SidH in *Legionella pneumophila* Paris strain. a** Schematic representation of split SidH gene- *lpp2883* encodes for C-terminal (SidH$^{Paris}$ C-term) while *lpp2886* encodes for N-terminal (SidH$^{Paris}$ C-term) of SidH from strain Paris. *lpp2884* and *2885* are the two transposons. **b** Analytical SEC shows interaction between N-terminal and C-terminal of SidH$^{Paris}$. The highlighted fraction (green horizontal bar) of SidH$^{Paris(N-term + C-term)}$ (blue colored curve) sample is shown on SDS-PAGE (right panel). **c** Cytotoxicity assay in HEK293T cells shows toxicity levels of different constructs of SidH$^{Paris}$ relative to the toxicity of SidH FL from Philadelphia strain. Data represents the mean ± SD of three independent reactions. **d** Analytical SEC shows that SidH$^{Paris}$ N-term apo protein has more affinity for pre-formed complex of EF-Tu/tRNA. Fractions of peak1, and 2 from each chromatogram were loaded onto SDS-PAGE (right panel). Source data are provided as a Source Data file for (**b-d**).

responsible for the proper localization of the protein by binding to specific host proteins or membranes while the N-terminus forms the functional core of SidH. We next investigated the nature of SidH interaction with t-RNA and EF-Tu. For this, we used the fraction of SidH$^{Paris}$ N-term that was purified in apo form (Fig. S9B). We incubated SidH$^{Paris}$ N-term with in vitro transcribed human tRNA-Ser-TGA or/and purified *E. coli* EF-Tu before subjecting the mixture to analytical size exclusion chromatography. Most of the SidH$^{Paris}$ N-term eluted in complex with tRNA and EF-Tu/GTP; while individually, tRNA and EF-Tu could not complex with SidH$^{Paris}$ N-term (Fig. 4d). This data indicates that a prior complex formation of tRNA and EF-Tu is necessary for maximum affinity towards SidH, with SidH exhibiting lower affinities for tRNA alone and EF-Tu alone.

**Functional implications of SidH-EF-Tu/tRNA complex structure**

In the cryo-EM structure described here, *LP* SidH is bound to *E. coli* tRNA and EF-Tu molecules. To put this finding in the context of *LP* infection, we sought to test if SidH binds to tRNA/EF-Tu in human cells. We used HEK293T cells for this purpose due to their high transfection efficiency and because both the tRNA and EF-Tu homologues are part of the core translational machinery that are invariant across cell types. We immunoprecipitated (IPed) the GFP-tagged SidH from HEK293T cells and employed quantitative proteomics to analyze the co-IPed proteins (Fig. 5a, Supplementary Data 1). We found most ribosomal proteins, several tRNA ligases being enriched along with SidH. Apart from these, other RNA-binding proteins were also enriched especially YBX-1 which is highly

enriched together with SidH. YBX-1 is an RNA-binding protein involved in sorting of RNAs including tRNAs into exosomes[31]. We however did not find EEF1A1 or TUFM being enriched with SidH in our mass spectrometry analysis. Overall, our data point to a translational or other RNA regulatory role for SidH consistent with its interaction with tRNA in vitro.

To further probe the functional relevance of the SidH and bacterial tRNA-EF-Tu interaction, we asked if the t-RNA binding mutations in SidH affect its toxicity in HEK cells. We expressed WT SidH Philadelphia and the tRNA-binding deficient SidH mutant (SidH HM K57E, K71E, K110E, A117E, K504E and R819E) in HEK cells. Expression of WT SidH Philadelphia led to rounding of cells and decreased cell density indicating toxicity whereas the SidH HM expression almost abolished this phenotype (Fig. S12B). To further quantify this phenotype, we assayed for cell death using the ToxiLight assay kit (Lonza) (Fig. 5b, Fig. S12C). This analysis revealed that the tRNA-binding deficient mutant SidH HM shows no toxicity when expressed in HEK cells. Expression of the SidH/EF-Tu interface mutant (SidH Triple Mutant (TM): W438A, V486E and S499E) in HEK cells also does not show any toxicity. Accordingly, expressing SidH with mutations disallowing binding to both tRNA and EF-Tu (SidH Octa Mutant (OM): K57E, K71E, K110E, A117E, K504E, R819E, W438A, and S499E) also showed no toxicity (Fig. 5b). In a different toxicity assay, we expressed WT SidH and a mutant of SidH where only three of the tRNA-binding residues are mutated (K71A, A117E, R819E) and monitored the growth of HEK cells using live microscopy (Fig. S12D–F and supplementary movies 1–3). This showed that these three t-RNA binding mutations

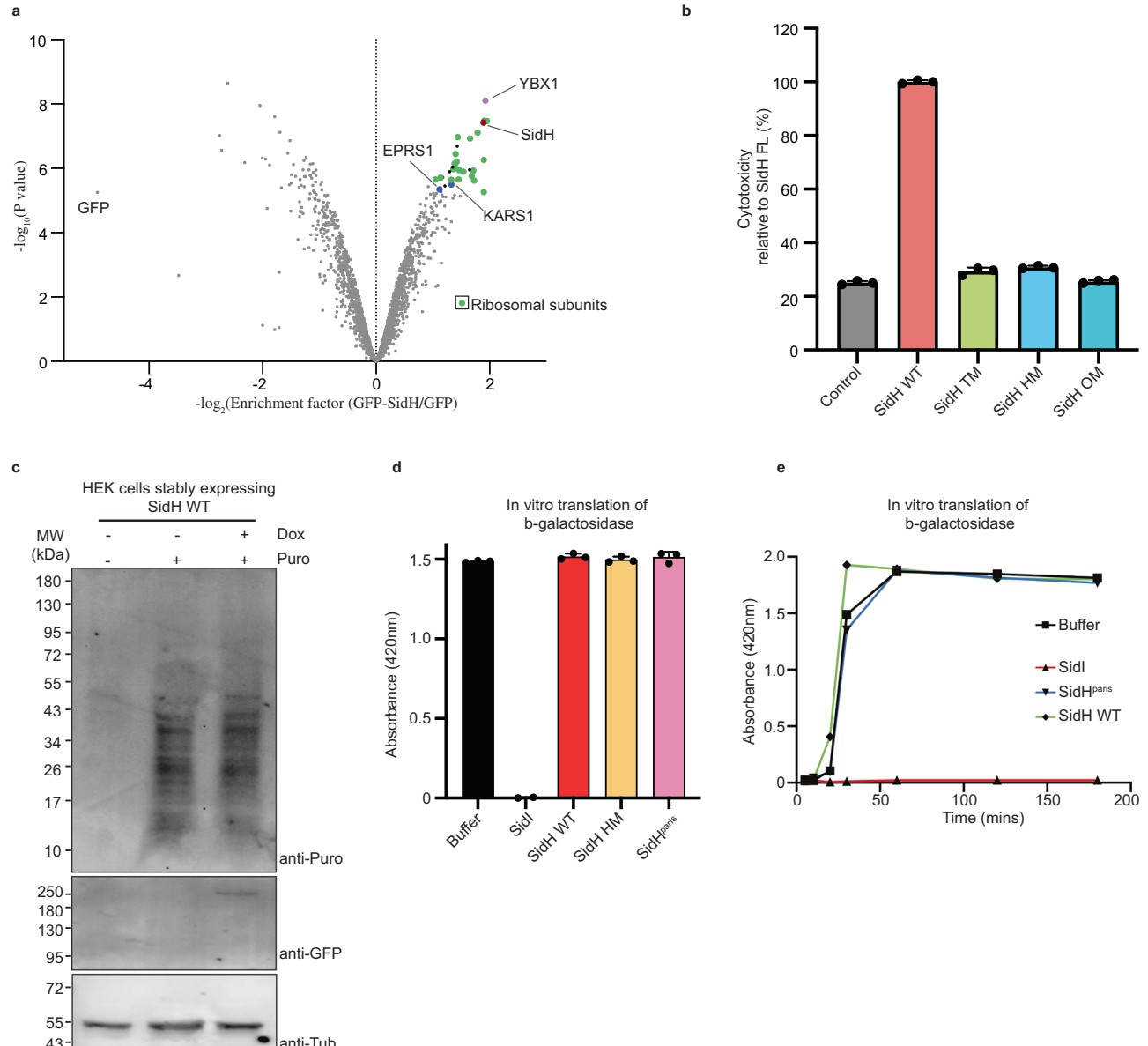

**Fig. 5 | SidH does not affect host protein synthesis. a** GFP-SidH or GFP alone was expressed in HEK293T cells and isolated proteins were subjected to quantitative mass spectrometry analysis. A volcano plot is shown here for the GFP-SidH inter-action proteomics. $n = 3$ biologically independent experiments. A protein was annotated as a hit with a false discovery rate (fdr) smaller than 5 % and a fold-change of at least 100% and as a candidate with a fdr below 20 % and a fold-change of at least 50 %. **b** The quantification of cell death shows that TM, HM and OM are not toxic in comparison to the SidH WT. Data represents the mean ± SD of three independent reactions. (SidH WT represents SidH from strain Philadelphia in the Fig. 5b–e) **c** HEK Cells stably expressing GFP-SidH WT subjected to puromycin incorporation analyses. Doxycyclin was used to induce the expression of GFP-SidH WT. Immunoblotting of lysates was performed using antibodies against puromycin, GFP and Tubulin. **d** Cell free translation of $\beta$-galactosidase in human cell extracts in the presence of purified His-tagged SidI/SidH WT/SidH HM/SidH$^{Paris}$. The reactions were carried out for 30 min. Graph depicts the activity of synthesized $\beta$-galactosi-dase on its substrate ONPG. Data represents mean ± standard error of the mean; $n = 3$ replicates per treatment condition. **e** Cell free translation of $\beta$-galactosidase in the presence of different proteins shows that SidH (both from Paris and Philadel-phia strains) is unable to repress the host protein synthesis at different time points. Source data are provided as a Source Data file for (**b**–**e**).

lead to near complete abolition of SidH-dependent inhibition of cell growth. Although these observations do not imply that SidH directly interacts with the human tRNA/EF-Tu, they indicate that tRNA/EF-Tu binding site in SidH is critical to the toxicity of SidH in HEK cells. It appears that SidH needs both the N- and C- termini to elicit toxicity in human cells consistent with the observations made with the SidH$^{Paris}$ proteins of strain Paris (Fig. 4).

Since tRNA and EF-Tu play a central role in protein synthesis, we asked whether SidH-induced toxicity is due to its effect on host protein synthesis. Towards this, we conducted puromycin incor-poration assays as an indication of active translation in HEK cells

stably expressing WT SidH Philadelphia under a tetracyclin-inducible promoter. Surprisingly, protein synthesis seemed unaf-fected upon SidH expression indicating that binding of tRNA by SidH does not lead to a notable global reduction in protein synthesis in HEK cells at least under the conditions tested here (Fig. 5c). We also used the human cell-free protein expression system (TAKARA) and tested the effect of adding purified WT SidH Philadelphia and the tRNA-binding mutant SidH HM on the in vitro synthesis of β-Galactosidase protein. Consistent with the observations made using the puromycin incorporation assays in HEK cells, the presence of SidH had no appreciable impact on protein synthesis in vitro

(Fig. 5d, e). To exclude the possibility that the pre-bound tRNA and EF-Tu might prevent *E. coli* purified SidH Philadelphia from affecting the translation, we used a complex of apo SidH[Paris] N-term and SidH[Paris] C-term to incubate with the in vitro translation assay components. Apo SidH[Paris] complex also failed to show any effect on protein synthesis in vitro (Fig. 5d, e). As expected, the potent translational repressor, the *LP* effector SidI, completely abolished the synthesis of β-Galactosidase under the same conditions[32]. These data indicate that although SidH binding of tRNA is toxic to cells, the toxicity is not a result of global attenuation of translation. We cannot however exclude the possibility that SidH is affecting local translation or translation of specific mRNA transcripts in cells.

### Structure of SidH-LubX complex

LubX is a *LP* metaeffector that ubiquitinates SidH in a temporal manner during *LP* infection[18]. The likely role of LubX in infection is to curb the toxicity of SidH after the initial infection period[33]. The crystal structure of LubX in complex with the ubiquitin-conjugating enzyme UBE2D2 was determined previously, but there is no structural information on the complex between SidH and LubX[34]. To gain insights into the regulation of SidH, we sought to determine the structure of SidH in complex with LubX. We co-expressed SidH and LubX in *E. coli* and purified the complex using affinity chromatography followed by size-exclusion chromatography (Fig. S13A, B). We performed Gradient Fixation (GraFix) to stabilize the complex of SidH-LubX before plunge freezing the sample in liquid ethane for single particle cryo electron microscopy analysis (Fig. S13C). 3D Focused classification in Relion helped us get the map of SidH bound to LubX with an overall nominal resolution of 3.1 Å (Fig. S14). The full length LubX crystal structure (PDB:4WZ3, chain B) and the SidH-EFTu/tRNA complex structure could readily be morphed into the SidH-LubX EM map. We then real-space refined the structure against the EM map in PHENIX[35] which resulted in the model of the SidH-EFTu/tRNA-LubX complex structure (Fig. 6a)[34]. SidH does not undergo even minor conformational changes upon binding to LubX. LubX bound to UBE2D2 and LubX in complex with SidH superposes well overall, with an r.m.s.d of 1.1 Å over 196 C-α's. Interestingly, the C-terminal helix of LubX which was unresolved previously in the crystal structure of LubX-UBE2D2, now becomes resolved when bound to SidH (Fig. S15A). The SidH and LubX interaction buries a surface area of 487 Å² that is mainly composed of polar interactions. Helical bundle 4 (Hb4) of SidH, which lies on the opposite side relative to the binding site of the EF-Tu/tRNA complex, mediates interaction with LubX (Fig. 6b). This implies that LubX can bind to and ubiquitinate SidH that is complexed with tRNA and EF-Tu. Accordingly, LubX was able to ubiquitinate both the EF-Tu/t-RNA complexed WT SidH and SidH HM which is devoid of these factors (Fig. S15B). Consistent with the previous biochemical analysis[34], the C-terminal Ubox2 domain of LubX lies in proximity to the SidH molecule (Fig. 6b). The C-terminal helix of LubX and the connecting helix of LubX which joins the two Ubox domains of LubX, primarily mediate interaction with the Hb4 of SidH (Fig. 6b). Specific interactions between SidH and LubX include Arg115, Arg119 from the connecting helix of LubX and Arg197 from the C-terminal helix of LubX forming polar interactions with SidH residues Asp684, Asp688, Thr691 and Asp694 (Fig. 6c). SidH and LubX also share a small hydrophobic interface involving residues Val190, Phe193 from the C-terminal helix of LubX and Leu566, the aliphatic part of Lys695 from SidH Hb4 (Fig. 6c). Interestingly, Arg119 of LubX was previously identified to be important for binding to SidH using large scale mutagenesis of LubX surface residues followed by functional screening in yeast[34]. LubX specifically seem to target SidH during infection even though other effectors of the SdhA family, SdhA and SdhB, share high sequence homology with SidH in the N-terminal 700 residues (Fig. S16). This can be explained by the lack of conservation of LubX-binding residues of SidH amongst the homologues SdhA and SdhB (Fig. S16).

Ubox1 which is the catalytic domain of LubX is not involved in interaction with SidH as previously reported[18,34]. From the local resolution analysis of the SidH-LubX cryo-EM map, we could deduce that the Ubox1 domain is probably flexible relative to the rest of the molecule (Fig. 6d). Based on the crystal structure of LubX bound to the E2-enzyme UBE2D2, we modeled the E2 into the SidH/LubX structure (Fig. 6e). To identify the lysines of SidH targeted for ubiquitination by LubX, we performed an in vitro ubiquitination assay and specifically analyzed ubiquitinated fraction of SidH using mass spectrometry. We found that SidH residues Lys230 in Hb2, Lys358, Lys369 in Hb2-Hb3 connecting loop and Lys656 in Hb4 are getting ubiquitinated (Fig. 6e, Fig. S17A, Supplementary Data 2). All these residues are present in loops and disordered but importantly lie near to the modeled E2 (Fig. 6e). Our mass spectrometry data also revealed that Lys6, Lys11 and Lys48 are the chains prevalent in the ubiquitinated form of SidH consistent with the role of LubX in proteasomal degradation of SidH (Supplementary Data 2). Next, we performed structure-based point mutations and created a SidH quadruple mutant (QM: L566D_D684A_D688A_T691D) to test if this SidH mutant would still be toxic in HEK cells but be resistant to LubX-mediated degradation. We transiently overexpressed SidH[Paris or Philadelphia] WT or SidH[Paris or Philadelphia] QM with or without LubX (Fig. S17). Expression of SidH[Paris or Philadelphia] QM showed similar toxicity in HEK cells to that of SidH[Paris/Philadelphia] WT but the toxicity of SidH[Paris/Philadelphia] QM which is resistant to the presence of LubX is in line with the prediction that LubX fails to bind and ubiquitinate SidH[Paris/Philadelphia] QM (Fig. 6f, Fig. S17B).

### Implications of SidH interactions with tRNA, EF-Tu and LubX in *LP* infection

To test the relevance of the binding of SidH to tRNA and EF-Tu in infection conditions, we constructed Δlpp2886 *L. pneumophila* Paris strain lacking the SidH[Paris] N-term. For assaying the intracellular growth of the bacteria, we infected *Acanthamoeba castellani*, the natural host of *Legionella* and lysed the cells and measured the bacterial growth by counting the colony forming units (CFUs) at various time points. The Δlpp2886 strain did not exhibit any growth defect compared to the *L. pneumophila* Paris wild-type strain (Fig. S18A). We next complemented the Δlpp2886 strain with a plasmid overexpressing (OE) wild-type SidH[Paris] N-term to see if the excess toxicity of SidH affects the growth of the bacteria. As expected, SidH overexpression led to several fold reduction in intracellular growth of the bacteria compared to the Δlpp2886 strain (Fig. 7a). We next complemented *the* Δlpp2886 strain (Paris) with a plasmid overexpressing SidH[Paris] N-term mutant that is deficient in binding to either tRNA or EF-Tu or LubX (Fig. 7b, Fig. S18B). Intracellular replication of these *LP* Paris strains was compared to determine whether the mutations have the same effect in infection as seen in vitro. In line with our structural and in vitro toxicity data, the overexpression of LubX binding deficient SidH mutant led to further growth delays as compared to the overexpression of SidH WT presumably because in the absence of LubX-mediated degradation, persistent toxic SidH levels are present in the host cell. In contrast, the OE of tRNA/EF-Tu binding deficient mutants of SidH led to more bacterial growth compared to OE of SidH WT confirming that SidH exerts its toxicity via the EF-Tu/tRNA-binding regions identified in the cryo-EM structure. In summary, SidH overexpression leads to toxicity in hosts, SidH resistant to LubX-mediated degradation increases its toxicity in host cells but removing tRNA or EF-Tu binding reduces its toxic phenotype.

### Discussion

A large number of *LP* effectors are toxic to eukaryotic cells when ectopically overexpressed, perhaps that is why, *LP* has evolved to contain so called metaeffectors that the bacteria employ typically during the later stages of infection to curb the excess toxicity of other effectors[17,22,23]. We recently characterized the effector-metaeffector

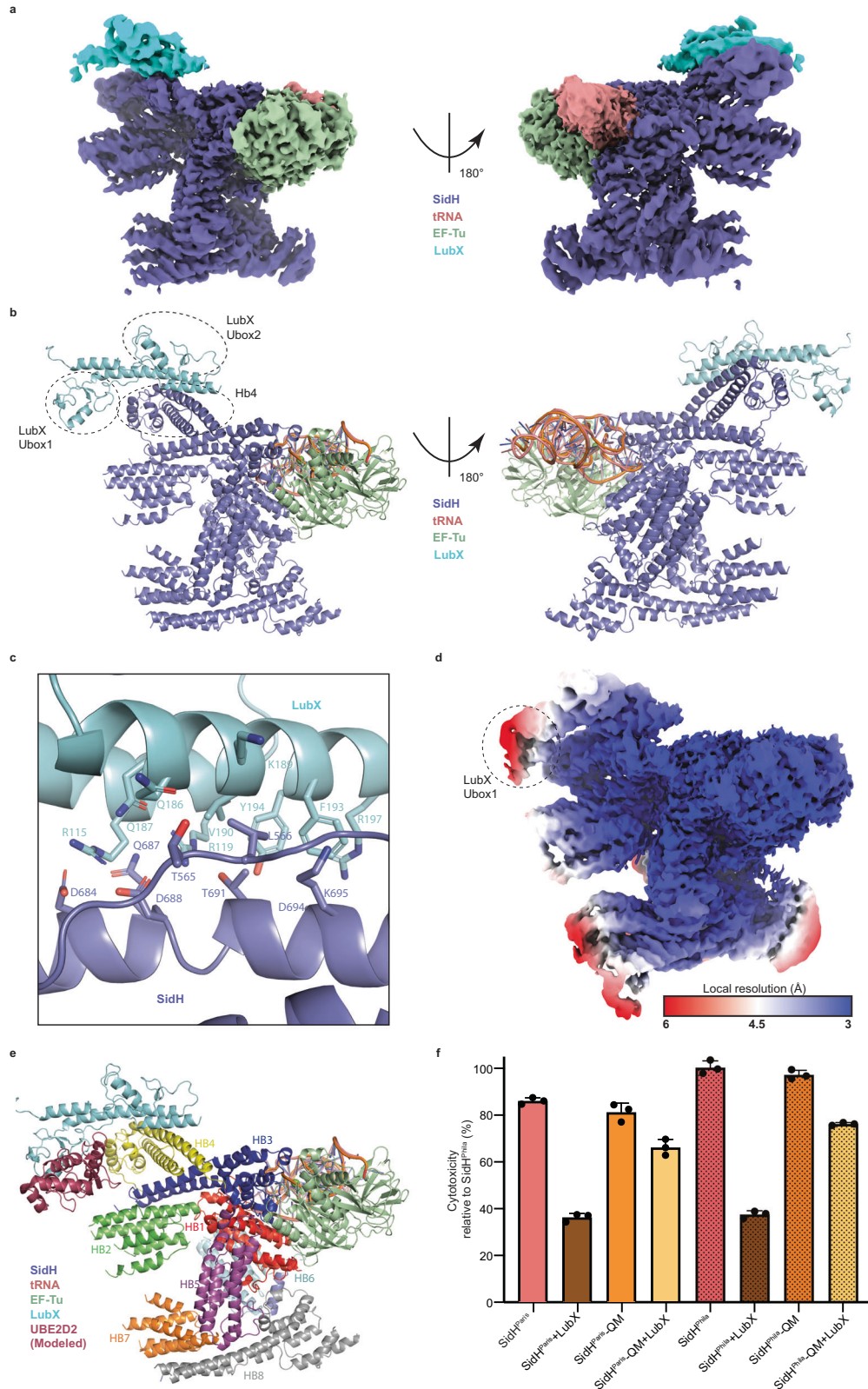

**Fig. 6 | LubX binds to N-terminal of SidH. a** The EM map for LubX in complex with SidH/EF-Tu/tRNA. The density for LubX is shown in Cyan. The rest of the color coding is same as in Fig. 1. **b** The model for the complex shows that Ubox2 of LubX interacts with Hb4 of SidH. **c** The detailed view of SidH-LubX interaction interface shows stick representation of key residues involved in the interaction. **d** Local resolution cryo-EM map shows poor electron density for the Ubox1 of LubX

(circled). **e** Modeled UBE2D2 (E2-enzyme) lies near the Hb2 of SidH. **f** The cytotoxicity assay with SidH or SidH QM and LubX constructs expressed in HEK293T cells validates the interaction surface between SidH and LubX. Data represents the mean ± SD of three independent reactions. Source data are provided as a Source Data file.

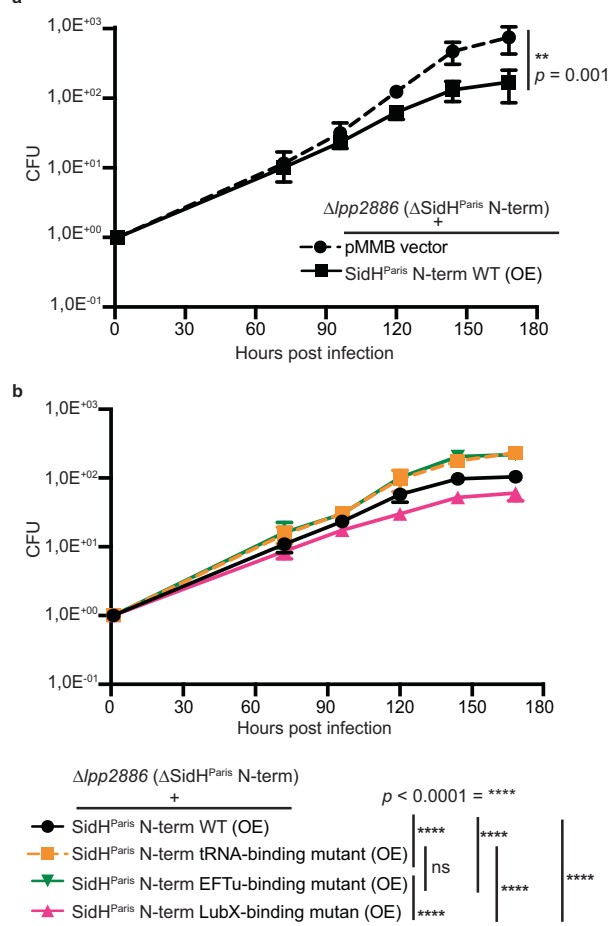

**Fig. 7 | SidH, tRNA/EF-Tu binding is relevant in infection.** Intracellular growth kinetics of *L. pneumophila* in *Acanthamoeba castellani*. The number of viable bacteria within amoebae was evaluated by the standard plate count assay. Each time point represents the mean of three biological replicates. Infections were performed at 20 °C. **a** Infection of *Acanthamoeba castellani* with Δ*lpp2886 L. pneumophila* Paris strain complemented with empty pMMB vector or with the one expressing SidH[Paris] N-term WT. **b** Infection of *Acanthamoeba castellani* with Δ*lpp2886 L. pneumophila* Paris strain complemented with the indicated mutants (black- WT; yellow- tRNA-binding mutant; green- EFTu-binding mutant and pink- LubX-binding mutant). Data presented here in Fig. 7 is represented by mean ± SD of three independent replicates. Statistical analysis used two-way ANOVA, with reported P values for significant comparisons. ($p < 0.0001$ - ****), ns- non-significant. Source data are provided as a Source Data file.

pair SdeA-SidJ in which SdeA is a toxic effector which performs non-canonical ubiquitination and toxifies host ubiquitin and SidJ is a glutamylase that silences the activity of SdeA[36,37]. Similarly, *LP* effector SidH is a toxin that is regulated by the U-box domain-containing ubiquitin ligase and metaeffector LubX.

The cryo-EM structure of SidH described here revealed that SidH adopts a remarkably unique structure with eight distinct bundles of helices packed in a unique fashion (Fig. 1). The structure also revealed that SidH interacts specifically with t-RNA and EF-Tu that are co-purified with SidH from *E. coli*. Sequencing of the RNA co-purified with SidH revealed that SidH does not specifically interact with a single tRNA but binds to all tRNA isotypes from *E.coli*. The conserved T-loop situated at the pivot of the L-shaped tRNA molecule sits in a positively charged groove on the surface of SidH. The acceptor arm of the tRNA is charged with an amino acid and is located in the active site of EF-Tu. The anticodon arm of tRNA which typically binds to the mRNA at the A-site of the translating ribosomes is projected away from the SidH-

tRNA-EF-Tu complex and into the solvent (Fig. 2). Since tRNA and EF-Tu of *LP* share high sequence similarity to the *E.coli* counterparts, it is likely that the endogenous SidH within *LP* binds to LP tRNA and EF-Tu. Intriguingly, the secretion signal of SidH has been mapped to the extreme C-terminus of the protein[38] which is distant from tRNA and EF-Tu binding sites and can participate in interaction with the subunits of the secretion system and undergo unfolding followed by translocation of SidH. Sequence analysis indicated that some residues involved in binding to SidH are conserved between EF-Tu and human elongation factors EEF1A1 and TUFM (Fig. S19, 20) indicating that binding between SidH and these human factors is plausible. SidH even bound to in vitro transcribed human tRNA in our analytical size-exclusion chromatography analysis (Fig. 4d). To test the binding of elongation factors or other translation associated proteins to SidH directly in human cells, we immunoprecipitated overexpressed SidH from HEK293T cells and analyzed the bound protein fraction using quantitative mass spectrometry. This revealed that SidH is complexed with a milieu consisting of ribosomal proteins, several tRNA ligases and other RNA-binding proteins but no human elongation factor proteins were found to be co-eluting with SidH. This raises the limitation of the current study in which we have not revealed the precise cellular or physiological context of the binding of SidH to *E.coli* tRNA/EF-Tu complex and the question that whether SidH binds to similar factors directly in the host cell cytoplasm still remains open.

It should be noted that the t-RNA and EF-Tu binding seems to be still relevant for the toxicity of SidH as mutants of SidH designed to disrupt SidH-tRNA and SidH-EF-Tu interfaces rendered SidH non-toxic in HEK cells as well as in infection (Figs. 5b, 7). However, SidH was not able to inhibit global protein synthesis in HEK cells as well as in vitro (Fig. 5c–e). It is possible that SidH is involved in the inhibition of local translational events which our assay conditions would be unable to probe. This is also suggested by our observation that SidH requires both the N-terminal and the C-terminal halves for its host toxicity even though t-RNA and EF-Tu binding regions are located in the N-terminus (Figs. 1f, 4c). Future experiments designed to monitor polypeptide synthesis in live cells with and without SidH will be able to address if SidH plays a role in protein translation. It is also possible that SidH-mediated toxicity is due to the binding of an unknown cellular RNA to the tRNA-binding site observed in the SidH structure. Interestingly, a recent study by Sahr et al. has shown that *LP* shed extracellular vesicles containing small RNAs including many different types of tRNAs that regulate the transcription of various host innate immune factors[4]. It is also possible that SidH binds to these small RNAs of *LP* secreted into the host during infection. Future experiments such as cross-linking, and immunoprecipitation (CLIP) aimed to identify the RNAs bound to SidH during infection will shed light on this.

Structure of LubX bound SidH revealed that LubX recognizes Hb4 of SidH which lies in the N-terminal region of SidH but distant from the binding surface of tRNA and EF-Tu (Fig. 6). This means that LubX can bind to SidH-EF-Tu-tRNA trimeric complex and target SidH for ubiquitination and degradation. In agreement with the previous reports, we found that the C-terminal Ubox2 domain of LubX interacts with SidH. Specifically, the connecting helix that lies in between the two Ubox domains and C-terminal helix of LubX are involved in the interactions with SidH through mostly ionic interactions (Fig. 6c). Based on the structure, we were able to design a LubX-resistant SidH which is as toxic as the wild type SidH but is more toxic to the *LP* natural host *A. castellini* in our infection experiment (Figs. 6f, 7). This is consistent with the idea that LubX-mediated temporal degradation of SidH is to suppress the persistent toxicity of SidH.

SidH is part of SdhA family of *LP* effectors which include SdhA and SdhB in addition to SidH. All three proteins of the SdhA family of effectors share 30–40% sequence similarity in the N-terminal ~800 residues. Despite this, the SidH residues involved in the binding of tRNA and EF-Tu are not conserved in SdhA and SdhB (Fig. S16)

indicating that the SdhA family proteins might perform different functions despite being classified as a single family of effectors. Accordingly it was found that SdhA is the only protein in the SdhA family of effectors that is essential for the intracellular replication of *LP* in primary macrophages[8]. Interestingly, SdhA has been previously implicated in cytosolic RNA-triggered INF response during *LP* infection[14]. Future studies exploring the structure and biochemistry of SdhA and SdhB will shed light on the exact mode of function of SdhA family of effectors and if the effectors SdhA and SdhB also bind RNA.

## Methods

### Cloning of SidH FL, SidH^paris^ N-term, SidH^paris^ C-term, LubX and their mutants

The full length SidH (SidH FL) was cloned into expression vector pCoofy18 with n-terminal 10xHis tag. The codon corresponding to Glycine1623 residue was modified to stop codon by site-directed mutagenesis (SDM) to obtain SidHΔtail construct. For SidHΔDUF, a region from 904 to 1238 was replaced with 12 residues long GSGS linker. SidH hexa mutant (HM) is a combination of 6-point mutations, which were introduced using SDM. Likewise, SidH triple mutant (TM) and SidH octa mutant (OM) were generated. GST-tagged LubX-expressing plasmid pGEX6P1-LubX was a kind gift from Prof. Hiroki Nagai (Osaka University, Japan). The *LP Paris* strain's genomic DNA (a gift from Prof. Carmen Buchrieser from Institute Pasteur, Paris), was used to amplify the *lpp2886* (corresponding to N-terminal half of SidH FL^Philadelphia^) and *lpp2883* (corresponding to C-terminal half of SidH FL^Philadelphia^) and cloned into pCoofy18 vector. For the SidH-LubX co-expression, the His-tagged SidH-expressing plasmid (pET15b-SidH was a gift by Prof. Hiroki Nagai), while LubX was cloned using SLIC cloning from Prof. Nagai's pGEX6P1-LubX plasmid into pCoofy1 with the introduction of a 3 C cleavage site. The cDNA sequence *(lpg 2504)* encoding SidI effector protein was chemically synthesized in pUC18 vector from GenScript and sub-cloned into pCoofy18 vector. Point mutations were introduced via site-directed mutagenesis using a custom primer pair (as described in Supplementary Table 2).

### Purification of SidH FL, SidH^paris^ N-term, SidH^paris^ C-term, LubX and their mutants

The 10x His tagged SidH FL was used to transform chemically competent BL21Star (DE3) (Sigma Aldrich). The cells were grown in Luria Broth (CondaLab) at 37 °C until OD600 reaches to 0.8. After a brief (30–60 min) incubation at 4 °C, the culture was then induced using 0.5 mM IPTG, and grown further at 18 °C for 18 h. Post-expression, the cells were harvested and the cell pellet was resuspended in lysis buffer (300 mM NaCl, 50 mM Tris pH 7.5, 10% glycerol) supplemented with 1 mM PMSF and the protease inhibitor cocktail (Roche). The cells were lysed by sonication followed by centrifugation at 39191 *g* for 45 min. The filtered supernatant was applied to 3 mL of Talon metal affinity bead resin (Takara) pre-equilibrated with lysis buffer and incubated via rolling at 4 °C for 90 min and centrifuged at 500 *g* for 3 min. The supernatant was removed, and the beads were washed twice with 10 column volume of lysis buffer. Additional washes in the presence of 5- and 10-mM imidazole and 800 mM NaCl were applied to remove more impurities. The 10xHis-SidH FL was eluted using elution buffer (300 mM imidazole, 10% Glycerol, 50 mM NaCl, 50 mM Tris pH 7.5). The eluted protein was loaded onto HiTrap Q HP anion exchange chromatography column (GE Life Sciences) pre-equilibrated in low salt buffer (10% Glycerol, 50 mM NaCl, 50 mM Tris pH 7.5) at the speed of 1 mL/min. The bound protein was eluted using a linear salt gradient from 50 mM NaCl to 1000 mM NaCl. All the peak fractions were checked for the presence of pure SidH FL protein using SDS-PAGE and the fraction containing SidH FL was concentrated and loaded onto a pre-equilibrated in size-exclusion chromatography (SEC) buffer (10 mM HEPES pH 7.5, 100 mM NaCl and 0.5 mM TCEP) Superdex 200

increase 10/300 size-exclusion column (GE Life Sciences). Peak fractions were evaluated via SDS-PAGE and the purest fractions were pooled, concentrated and the concentration was measured using Micro BCA Protein Assay Kit (Thermo Scientific). The Hexa Mutant of SidH, SidH^paris^ N-term, SidH^paris^ C-term and SidI were purified using the same protocol.

For the SidH-LubX complex, the two proteins were co-transformed in BL21Star (DE3) (Sigma Aldrich). The cells were grown in Luria Broth supplemented with Amp/Kan/Chlo at 37 °C until OD600 = 0.6. After cooling the cells, expression was induced using a final IPTG concentration of 0.5 mM. After induction at 18 degrees for 18 h, the cells from 8 l of liquid culture were harvested via centrifugation and the pellet resuspended in 35 mL lysis buffer containing 300 mM NaCl, 50 mM Tris pH 7.5, 10% Glycerol, supplemented with a protease inhibitor cocktail tablet (Roche). Lysis of the pellets was performed via sonication, after which the lysate was clarified via centrifugation at 39191 *g* for 45 min and filtered using a 0.22 μM syringe filter, before applying the lysate to 4 mL of pre-equilibrated talon resin (Takara). After 1 h of gentle rolling incubation at 4 °C, the flow-through was decanted by centrifuging the slurry at 500 g for 2 min. The pellet was washed 3 times with 40 mL lysis buffer, incubated for 10 min, and centrifuged at 500 g for 2 min to decant the liquid. Then, the protein was washed one last time with 25 mL lysis buffer supplemented with 10 mM Imidazole, before eluting the protein using 3 elution steps with 10 mL lysis buffer supplemented with 100 mM, 200 mM, and 300 mM Imidazole. Using SDS-PAGE, the sample purity was evaluated and the 100 mM elution fraction was chosen for further purification. The filtered lysate was concentrated and injected onto a Superdex S200 increase 10/300 (GE Life Sciences) column pre-equilibrated with SEC buffer (10 mM HEPES pH 7.5, 100 mM NaCl, 0.5 mM TCEP). The resulting fractions were evaluated via SDS-PAGE and the fraction of highest purity was used for GraFix cross-linking.

### GraFix cross-linking

After purification of SidH-LubX, 280 uL of sample at a concentration of 0.4 mg/ml was applied to glycerol gradients containing 5–25% glycerol and 0–0.1% glutaraldehyde, and subsequently ultracentrifuged using an SW60Ti rotor at 40.000 rpm for 18 h. The gradients were then quenched using 20 mM (final concentration) Tris pH 7.5 by fractionating 300 uL fractions into wells containing 1 M Tris. Samples of each fraction were evaluated via SDS-PAGE and silver staining, and desired fractions were pooled. Excess glycerol was removed adding the pooled fractions to a 0.5 mL Amicon Ultra 100kDA MWCO spin concentration column (Merck Millipore) and centrifuging at 4.000 g for 10 min, after which the flow-through was removed and 400 uL of SEC buffer added to the sample. This process was repeated 5 times before the sample was finally concentrated to a target concentration of 0.3 mg/ml for grid preparation.

### tRNA sequencing

SidH FL purified from *E. coli* expression host was used for the tRNA extraction. Equal volume of purified protein was mixed with Phenol/chloroform/Isoamylalcohol (25:24:1) (SIGMA; Cat.#77617) solution, vortexed and centrifuged at 18,407 *g* for 15 min at RT. The top aqueous phase collected and mixed with 3 M Sodium Acetate pH 5.2 and ethanol. This mix was incubated at −20 °C for one hour and centrifuged. The resulting pellet was washed with 80% ethanol and dried. Finally, ~4 μg of extracted tRNA was processed at Genecore facility at EMBL Heidelberg for Small-RNA sequencing. The sequencing reads were processed (adapter sequence removal, trimming of 5'and 3'sequence, filtering out the sequences of less than 15 bases). The resulting 18117097 reads were aligned using bowtie2 tool against *E. coli* genomic and *E. coli* tRNA index which showed the overall alignment rate of 94.2% and 92.7% respectively. The GraphPad Prism was used to generate a pie-chart containing different percentage of tRNA isotypes.

## Peptide Mass-fingerprinting

*E. coli* purified SidH was loaded onto SDS-PAGE and the bands running at 100 kDa and 46 kDa molecular weight were cut and deposited at proteomics Core Facility at EMBL Heidelberg for in-gel acid hydrolysis. The samples prepared in reconstitution buffer (96:4 water: acetonitrile, 1% formic acid) subjected to LC-MS/MS analysis. An UltiMate 3000 RSLC nano LC system (Dionex) fitted with a trapping cartridge and an analytical column was used. The outlet of the analytical column was coupled directly to a Orbitrap Fusion Lumos (Thermo Scientific, SanJose) mass spectrometer using the nanoFlex source. The instrument was operated in data dependent acquisition (DDA) mode and MSMS scans were acquired in the Iontrap in rapid mode. IsobarQuant and Mascot (v2.2.07) were used to process the acquired data, which was searched against Uniprot *Escherichia coli* (UP000000625) proteome database containing the sequence of the expressed protein, common contaminants and reversed sequences.

## Mass-photometry, cryo-EM sample preparation

To prepare grids for cryo-EM data collection, purified SidH FL was first checked for the presence of any heterogeneity using mass photometry. First of all, the SEC buffer was used to determine the background. For calibration, we used Nativemark™ Unstained Protein Standard (LC0725-ThermoFisher Scientific) which contains known molecular weight marker proteins of 66, 146, 480, and 1048 kDa. Immediately prior to mass photometry measurements, protein stock was diluted directly in SEC buffer to 50 nM concentration. The diluted protein sample was applied onto the instrument, data was collected, processed and a graph was generated between counts and mass.

Quantifoil Au 300 1.2/1.3 grids were glow-discharged for 30 s using a Pelco EasyGlow on both sides at 30 mA current. SidH FL protein sample (2.5 µl) of 0.8 mg/ml was applied on each side to these freshly glow-discharged grids. They were then blotted inside a Vitrobot MkIV (Thermo Scientific) at 4 °C with 100% humidity and blot force −4 for 2 s and plunge frozen in liquid ethane cooled by liquid nitrogen.

For the SidH-LubX complex, Quantifoil Au 300 1.2/1.3 grids were glow-discharged in the same way. 3 uL of protein sample at a concentration of 0.3 mg/ml were applied to each side of the glow-discharged grid. Using a Vitrobot Mk IV (Thermo Scientific), the grids were then blotted at 4 °C with 100% humidity and a blot force of 0 for 2 s before being plunge frozen in liquid ethane.

## cryo-EM data acquisition

For the SidH FL, cryo-EM movies were collected on a Titan Krios (FEI) electron microscope equipped with a K2 Summit direct electron detector and a GIF Quantum energy filter (Gatan) at the EMBL Heidelberg cryo-EM platform. The operational acceleration voltage for the microscope was 300 kV. The defocus values ranged from −0.8 to −2.0 µm. The dose rate on the camera was set to be about 3 e⁻/pixel/second in electron counting mode. The total exposure time for each movie was 8 s with a total exposure dose of 41.93 e/ Å² on the sample. Each movie was fractionated into 40 frames, with 0.2 s per frame. 9020 images were collected with a pixel size of 0.81 Å.

For SidH_LubX complex, 9195 images were collected with a pixel size of 0.827 Å on Titan Krios (FEI) electron microscope at the ESRF Grenoble cryo-EM platform CM01 equipped with K2 direct electron detector[39]. The images were collected in electron counting mode with a dose rate of about 7 e⁻/pixel/second. The total exposure time for each movie was 4 s with a total exposure dose of 43.37 e/ Å² on the sample. Each movie was fractionated into 40 frames, with 0.1 s per frame.

## cryo-EM data processing

For SidH FL Cryo-EM data, motion correction and contrast-transfer function (CTF) estimation, with subsequent particle picking using the BoxNet2Mask_20180531 were performed in WARP. Coordinates of 1,370,015 particles were imported into Relion 3.1 and initially extracted

with a 2-fold binning factor. After two rounds of reference-free 2D classification 1,271,922 particles were included for 3D classification. One round of 3D classification gave rise to 6 different 3D classes. A subset of 367,741 particles from the best class were picked and re-extracted un-binned and 3D refined 4 times to 2.76 Å resolution. After post-processing, we had a map of 2.72 Å resolution. The reported overall resolution of 2.7 Å was calculated using the gold-standard Fourier shell correlation (FSC) 0.143 criterion.

For SidH-LubX complex, motion correction and contrast-transfer function (CTF) estimation, with subsequent particle picking using the BoxNet2Mask_20180918 were performed in WARP. Coordinates of 1,155,271 particles were imported into Relion 3.1[26] and initially extracted with a 4-fold binning factor. After one round of reference-free 2D classification, 894.096 particles were classified into 8 3D classes. A subset of 590.543 particles from the 6 classes were re-extracted with a binning factor of 4, refined, and local classification was performed. 255.829 particles from the 2 best classes were picked and refined to a final resolution of 3.06 Å.

## Ab initio model building and refinement

The crystallographic model of EF-Tu-tRNA^Phe (PDB 1TTT) structure was docked into the density. Iterative model building was performed both manually in Coot and using Map-to-Model in the Phenix package. Residues 903-1040 of SidH were modeled using the predicted Alphafold[25] structure which was docked into the low-resolution EM-density and appended to the rest of the model. The entire model of SidH-EF-Tu-tRNA-GTP was real space refined using PHENIX.

For SidH-LubX complex- PDB 4WZ3 of LubX/UbE2D2 and SidH model as initial models for rigid body fitting. LubX model was extended manually at the C-terminus. The entire model was real space refined using PHENIX[35].

## Cell lines

Doxycycline-inducible stable HEK Flp-In TRex (ThermoFisher SCIENTIFIC) GFP-SidH FL cell line was generated by transfection of Flp-In T-Rex cells with pcDNA5/FRT/TO GFP-SidH FL and Flp-recombinase pOG44. 48 h after the transfection, cells were then sub-cultured in a selection medium containing 15 µg/ml blasticidin and 100 µg/mL hygromycin. Resistant cell colonies were expanded and tested for doxycycline inducibility of the transgene. HEK293T cells (ATCC" CRL-3216™) were cultured in DMEM supplemented with 10% FBS at 37 °C, 5% CO₂. Cell lines were authenticated using STR DNA profiling. The cell lines used in the study are not in the commonly misidentified lines list.

## Puromycin incorporation assay

Stable cell line of HEK Flp-In TRex GFP-SidH FL was used to measure the effect of SidH on host protein synthesis. 1 µg/mL of doxycycline was used to induce the expression of SidH FL. 16 hrs after the induction, puromycin treatment was given to the cells at the concentration of 5 µg/mL for 15 min. The cells were washed with PBS and resuspended in a buffer containing 50 mM Tris pH 7.5, 150 mM NaCl, 1% Triton X-100 and protease inhibitor cocktail and lysed by incubating the lysate at 4 °C for 15 min. Western blotting method was used to transfer the proteins onto membrane and visualized them by incubating with GFP sc-9996 (Santa Cruz Biotechnology, dilution- 1:2000), alpha Tub DM1A NB100-690SS (Bio-Techne Ltd., dilution- 1:5000) and Puromycin 12D10 MABE343 (MERCK, dilution- 1:25,000) antibodies. Secondary antibodies IRDye® 800CW Donkey anti-Mouse IgG (Li-Cor, dilution-1:20,000) and IRDye® 680RW Donkey anti-Mouse IgG (Li-Cor, dilution-1:20,000) were used to visualize by fluorescence.

## Toxicity assay

The toxicity of different constructs of SidH was measured using ToxiLight™ Non-Destructive Cytotoxicity BioAssay kit (Lonza; Catalog #: LT17-217). N-terminal GFP tagged constructs (SidH FL/SidHΔtail/

SidHΔDUF/HM/TM/OM/SidH$^{paris}$ N-term/SidH$^{paris}$ C-term) cloned in mammalian expression vector pcDNA5 were used to transfect the HEK293T cells. 1 μg of each of these plasmids was mixed with transfection reagent polyethylenimine (PEI) in 1:3 ratio for the transfection. 20 h post-transfection, cells were imaged under the ZEISS Axio Vert.A1 microscope. To measure the toxicity, the supernatant DMEM media was collected and centrifuged at 300 g for 3 min at 4 °C to pellet down any remaining cells came along with the media while pipetting. After centrifugation, 20 μl supernatant was carefully removed from the top and added to the 100 μl of Adenylate Kinase detection reagent (AKDR) in 96-well plate provided with the kit. After 15 min of incubation at room temperature (RT), the total emitted light was measured using a Clariostar plate reader and graphs were prepared using GraphPad Prism. To detect the expression level of all SidH constructs, western blotting method was followed using 30 μg of the total protein lysate loaded onto Pre-cast 7.5% Mini-PROTEAN Tris-glycine gels (BIO-RAD).

SidH$^{paris}$ N-term was also cloned in HA-tagged mammalian expression vector. The LubX was cloned with n-terminal mCherry tag. These constructs were used with GFP-SidH$^{paris}$ C-term for toxicity assay. The Antibody used for mCherry detection was DSRed2 sc-101256 (Santa Cruz 692 Biotechnology), GFP was detected with GFP sc-9996 (Santa Cruz Biotechnology) and HA was detected with HA-tag C29F4 MAB3724 (Cell Signaling Technology). Secondary antibodies IRDye® 800CW Donkey anti-Mouse IgG (Li-Cor, dilution- 1:20,000), IRDye® 800CW Goat anti-Rabbit IgG (Li-Cor, dilution- 1:20,000) and IRDye® 680RW Donkey anti-Mouse IgG (Li-Cor, dilution- 1:20,000) were used to visualize by fluorescence.

### Imaging of HEK cells using live microscopy

HEK293T cells were purchased from ATCC and were used for transient transfections. These cells were maintained in a 37 °C incubator with humidified 5% CO$_2$ atmosphere and cultured in Dulbecco's modified Eagle's medium (DMEM) (Thermo Fisher Scientific, 31885049) supplemented with 10% foetal bovine serum (FBS) (Thermo Fisher Scientific, A5256801) and 100 U/ml penicillin and streptomycin (P/S) (Thermo Fisher Scientific, 15140130). For transient expression, DNA plasmids were transfected with JetPEI (Polyplus, 101000020) according to manufacturer instructions. In brief, 30 μl of transfection reagent were mixed with 10 μg of DNA and incubated for 20 min. HEK293T cells (2000 cells/well) were then plated in 384-well format simultaneously with addition of transfection complexes. Cells were incubated for 24 h, and subsequently imaged with IncuCyte ® S3 (Sartorius, Germany). Fluorescence of GFP as well as cell confluence (phase) were monitored over 48 to 72 h every 2 h. Each data point represents the averaged ratio of data obtained from three individual wells (technical replicates).

### Analytical gel filtration

To test the interaction between SidH$^{paris}$ N-term and SidH$^{paris}$ C-term, analytical gel filtrations were performed using 10 μM of both of these proteins. The sample was incubated on ice for 30 min before being injected into a pre-equilibrated (100 mM NaCl, 10 mM HEPES pH 7.5, 0.5 mM TCEP) Superdex SD200 3.2/300 size exclusion column (GE Life Sciences). The fractions were loaded on SDS-PAGE to visualize the protein bands. The resulting chromatograms were plotted using GraphPad Prism.

10 μM of apo SidH$^{paris}$ N-term was incubated with 10 μM of in vitro transcribed tRNA$^{Phe}$ (a kind help from Eva Kowalinski Lab) and 10 μM of E. coli purified EF-Tu in the presence of 1 mM GTP (Merck) and 2.5 mM MgCl$_2$ on ice for 30 min. The mix was loaded onto pre-equilibrated (100 mM NaCl, 10 mM Hepes pH 7.5, 2.5 mM MgCl$_2$) S200 3.2/300 size exclusion column.

### In-vitro translation assay

Human Cell-Free Protein Expression System from Takara (Cat. #3281) was used to determine the effect of SidH on host protein synthesis.

0.5 μM of purified proteins (SidH FL/SidI/HM/SidH$^{Paris}$) were mixed with different components of the kit including a plasmid which encodes for β-galactosidase in a 20 μl reaction. The reaction mix was incubated at 30 °C for different time points. 2 μl of the reaction mix was incubated with 0.7 M of o-nitrophenyl- β-D-galactopyranoside (ONPG) (MERCK; Cat. #73660) 37 °C for 30 min in cleavage buffer (60 mM Na$_2$HPO$_4$.7H$_2$O, 40 mM NaH$_2$PO$_4$.H$_2$O, 10 mM KCl, 1 mM MgSo$_4$.7H$_2$O, pH 7.0, 0.4 M β-Mercaptoethanol). The reaction was stopped by adding 0.5 M of Sodium Carbonate. Absorbance was measured at 420 nm using the plate reader and graphs were plotted. This experiment was done in triplicates.

### Strains, media, growth conditions and A. castellanii infection assay

Legionella. pneumophila Paris strains were cultured until reaching OD 4.2 in N-(2-acetamido)−2-aminoethanesulfonic acid (ACES)-buffered yeast extract broth or on ACES-buffered charcoal-yeast (BCYE) extract agar at 37 °C containing 10 μg/ml chloramphenicol and 1 mM IPTG. A. castellanii ATCC50739 was cultured in PYG 712 medium (2% proteose peptone, 0.1% yeast extract, 0.1 M glucose, 4 mM MgSO$_4$, 0.4 M CaCl$_2$, 0.1% sodium citrate dihydrate, 0.05 mM Fe(NH$_4$)$_2$(SO$_4$)$_2$ x 6H$_2$O, 2.5 mM NaH$_2$PO$_3$, 2.5 mM K$_2$HPO$_3$) at 20 °C. A. castellanii infection assays were performed as previously described in ref. 40.

Briefly, amoebae were infected with an MOI of 1 and incubated for one week at 20 °C. Intracellular multiplication was monitored daily by taking 350 μl from each experimental batch. Samples were then centrifuged (14,500 rpm, 3 min, rt) and vortexed for one minute to break up amoeba. After plating the bacteria on BCYE agar with chloramphenicol (10 μg/ml), the numbers of colony forming units (CFU) were determined after one-week incubation at 37 °C. Each infection was carried out at least in n = 3 biological repeats.

To detect expression of proteins, the corresponding Legionella pneumophila strains were grown until post-exponential phase, and 1 ml centrifuged at 5000 g. The pellet was resuspended in lysis buffer (20 mM Tris-HCl pH7.5, 150 mM NaCl, 1 mM EDTA, 1% NP-40, 1% Na-deoxycholate, 0.1% SDS, 5% glycerol) + protease inhibitor (Thermo Scientific) for total protein extraction by sonication. After treatment with benzonase (Sigma) and centrifugation, soluble total protein was quantified and an equal amount from each condition was spiked with loading buffer (4xLB: 200 mM Tris, pH6.8, 8% SDSD, 40% glycerol, 400 mM DTT, 0.01% bromphenol blue), denatured at 80 °C for 15 min and loaded on Criterion TGX Stain-free (4–15%). Separated proteins were transferred on 0.2 μM PVDF membranes using the TransBlot Turbo from Biorad. The expression of the HiBiT-SidH protein was detected using the Nano-Glo® HiBiT Blotting System from Promega. As a loading control the Anti-RNA polymerase beta antibody was used (Abcam, 1:1000) together with secondary antibody the HRP-linked Rabbit IgG (Cell Signaling, 1:5000). The Immunoblot was visualized with WesternBright Sirius HRP Substrate (Diagomics) and revealed using the G:Box system (Syngene) and the GeneSnap and GeneTools (Syngene) for analyses and quantification.

### Immuno precipitation and Mass spectrometry

To identify the interactors of SidH in human cells, GFP-SidH, and GFP-apo (control) plasmids were used for transient transfection in HEK293T cells in triplicates. Cells were harvested at 20 h post-transfection, resuspended in the lysis buffer (50 mM Tris pH 7.5, 150 mM NaCl, 1% Triton X-100, RNase inhibitor and protease inhibitor cocktail) and lysed by incubating it on ice for 15 min. The supernatant was collected after centrifugation and protein concentration was estimated using Micro BCA Protein Assay Kit (Thermo Scientific). 500 μg of the total protein was mixed with 20 μl of the GFP-Trap Agarose beads (ChromTek) and incubated for 2 h at 4 °C while being subjected to end-to-end rotation. The beads were washed three times with lysis buffer and then three times with wash buffer

(lysis buffer – triton X-100). Proteins were eluted by boiling with 2x Laemmli buffer for 10 min.

To obtain quantitative information, the samples were digested using trypsin (sequencing grade, Promega) and labelled with TMT10plex Isobaric Label Reagent (ThermoFisher). The fractionation was carried out on an Agilent 1200 Infinity high-performance liquid chromatography system, equipped with a Gemini C18 column (3 μm, 110 Å, 100 × 1.0 mm, Phenomenex). The peptides were introduced into the Fusion Lumos and data dependent acquisition was performed.

Data analysis was done with MSFragger v3.7[41] In brief, samples were searched against the Swissprot *Homo sapiens* database supplemented with GFP-apo and GFP-SidH sequence (IP samples) and eventually quantified by the TMT10plex option. Log2 transformed raw TMT reporter ion intensities were normalized. Proteins were tested for differential expression using the limma package[42]. A protein was annotated as a hit with a false discovery rate (fdr) smaller 5% and a fold-change of at least 100 % and as a candidate with a fdr below 20% and a fold-change of at least 50%.

### Ubiquitination assay

2 μM SidH WT or SidH HM, 1 μM UBE2D2, 1 μM GST-Lubx, 0.3 μM E1, 12 μM ubiquitin and 2.5 mM ATP were incubated for 1 h at 30 °C. Samples were loaded onto 4–20% Tris glycine gradient gels (Biorad) and analysed by Coomassie staining or western blot using either Ubiquitin P4D1 (sc-8017, SantaCruz) or GST (sc-138, SantaCruz) antibody in a solution containing 5% BSA, 0.2% Tween20 and PBS. Following numerous washing steps with 0.2% Tween20 in PBS, blots were incubated with IRDye® 800CW Donkey anti-Mouse IgG Secondary Antibody (Li-Cor) for 1 h at room temperature and visualized by fluorescence. Following SDS-P AGE and coomassie staining, the gel streak corresponding to ubiquitinated SidH were excised and subjected to in-gel digestion with trypsin to generate peptides containing the Lys-ε-Gly-Gly (diGLY) remnant. The peptides were introduced into the Orbitrap Fusion Lumos (Thermo Scientific, SanJose) mass spectrometer. Data analysis was done with MSFragger v3.7[41]. Carbamidomethyl (C) was set as fixed modification, Acetyl (Protein N-term), Oxidation (M) and GlyGly (K) as variable modifications.

### Statistics and reproducibility

No statistical method was used to predetermine sample size. No data were excluded from the analyses. The experiments were not randomized. The investigators were not blinded to allocation during experiments and outcome assessment. The experiments shown in Figs. 3e, 4b, 4d, and 5c of this publication have been performed at least three independent times with similar results.

### Reporting summary

Further information on research design is available in the Nature Portfolio Reporting Summary linked to this article.

## Data availability

Mass spectrometry data is available from the Proteomics Identification (PRIDE) database with the dataset identifier PXD044950. The RNA sequencing data is available from the ENA (European Nucleotide Archive) database with the dataset identifier ERR11775543. Cryo-EM structure coordinates are available from the PDB and the Electron Microscopy Data Bank (EMDB) for the SidH-EF-Tu-tRNA complex under accession codes 8QFS and EMD-18383, respectively. The coordinates and cryo-EM density for the LubX-SidH-EF-Tu-tRNA complex are available using accession codes 8QHC and EMD-18407. Models of tRNAphe-EF-Tu complex (PDB ID- 1TTT) and LubX-UBE2D2 (PDB: 4WZ3) were used in this study. Source data are provided with this paper.

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

## Acknowledgements

We thank the EMBL Mass spectrometry core facility, especially Mandy Rettel and Frank Stein for the data acquisition and analysis, Prof. Hiroki Nagai (Osaka University) for the kind gift of LubX and SidH expression plasmids, Martin Pelosse for training and maintenance of the EMBL eukaryotic expression facility, Eva Kowalinksi and her group for help with RNA sequencing data analysis and providing in vitro synthesized RNA and the cryo-EM platform of the Structural and Computational Biology Unit at EMBL Heidelberg. We thank the EMBL Genomic Core Facility for the RNA sequencing. We acknowledge the European Synchrotron Radiation Facility for provision of beam time on CM01. This work used the platforms of the Grenoble Instruct-ERIC center (ISBG; UAR 3518 CNRS-CEA-UGA-EMBL) within the Grenoble Partnership for Structural Biology (PSB), supported by FRISBI (ANR-10-INBS-0005-02) and GRAL, financed within the University Grenoble Alpes graduate school (Écoles Universitaires de Recherche) CBH-EUR-GS (ANR-17-EURE-0003). The work was supported by a grant from the French Agence Nationale de la Recherche (ANR-21-CE11-0013) to S.B. and by grants of the Fondation pour la Recherche Médicale (EQU201903007847) and the French Agence Nationale de la Recherche (ANR-10-LABX-62-IBEID) to C.B.

## Author contributions

R.S. performed protein purification, cryo-EM, biochemistry, cell toxicity assays and prepared samples for mass spectrometry. M.A. performed protein purification and cryo-EM of SidH-LubX complex. S.G.J. performed mutagenesis, in vitro ubiquitination and sample preparation for mass spectrometry. S.G. created the stable cell line for SidH and helped in other cellular assays. M.H. and F.W. collected the cryo-EM data. T.S. performed *L. pneumophila* infection experiments and LGV performed comparative genomics and phylogenetic analyses under the supervision of C.B.; R.B. performed toxicity assay using live microscopy under the supervision of A.S.; S.B. supervised the study and wrote the manuscript with contributions from all the authors.

## Funding

## Competing interests

The authors declare no competing interests.
