## [Peer Review File · Nature Communications]

Structural basis for the toxicity of *Legionella pneumophila* effector SidHREVIEWER COMMENTS

Reviewer #1 (Remarks to the Author):

The authors set to investigate the cellular function of the Legionella effector protein SidH, a paralog of an essential virulence factor SdhA, with which it shares ~40% sequence similarity over the N-terminal third of the protein. SidH is a 2225 amino acid long protein in *L. pneumophila* strain Philadelphia but in *L. pneumophila* strain Paris (and some other Legionella) strains the homolog of SidH is encoded by two genes, *lpp2883* and *lpp2886*. At the later stage of infection SidH is a target for another Legionella effector, LubX, which ubiquitinates SidH, leading to its degradation.

Recombinant SidH that was expressed in *E. coli* purified with a tightly bound two other macromolecules. They were identified by sequencing as an *E. coli* elongation factor EF-Tu and a single molecule of tRNA. Subsequently, the authors determined the structure of the SidH(1-1620)-EF-Tu-tRNA by cryo-EM at the nominal resolution of 2.7 Å. The structure is made of eight α -helical bundles with sequence and structural similarities between N- and C-terminal segments, suggesting a gene duplication event during evolution. The SidH is toxic to human cells and this effect is predominantly associated with the C-terminal unmodeled segment. The binding sites of EF-Tu and tRNA on SidH were determined and further experiments with the homolog from the Paris strain showed that the formation of the EF-Tu-tRNA complex is a prerequisite for binding to SidH. Moreover, tRNA interacts with SidH predominantly through its T-loop which is highly conserved across all tRNAs and explains why many different tRNAs sequences were found by sequencing of the purified complex.

Based on the association of SidH with the *E. coli*'s EF-Tu and tRNA, the authors hypothesize that during infection SidH binds to the human elongation factor EEF1A1 or mitochondrial TUFM, and tRNA. Unfortunately, the attempts to purify SidH from human cells were unsuccessful, likely due to high toxicity (authors suggestion). To confirm the importance of EF-Tu and tRNA binding to SidH for its toxicity, the authors showed that mutating the residues crucial for their binding, based on the structure, leads to a diminished toxicity. The potential of SidH affecting protein synthesis through its interaction with the elongation factor 1 α -1 was considered, however. additional experiments showed no such effect of protein synthesis.

To investigate the impact of LubX on SidH the authors determined the cryo-EM structure of this complex. LubX binds to SidH in the presence of bound EF-Tu and tRNA, at a site distant from the latter binding sites. Their modeling of UBE2D2 onto LubX based the crystal structure of their complex, suggested that the ubiquitination site is located on the helical bundle Hb2. Mutating SidH residues participating in LubX binding led, indeed, to prolonged toxicity of this mutant when co-expressed with LubX as compared to the wt SidH.

Overall, while the authors have not identified the specific mechanism of SidH toxicity, their structural findings identified association with EF-Tu and tRNA, showed that the binding of EF-Tu and tRNA is essential for toxicity, showed that LubX binds at a site distant from EF-Tu and tRNA and indicated the Hb2 region as the most likely site for ubiquitination. These are important discoveries of general interest.

I have several comments.

Regarding the potential interactions of SidH with eukaryotic elongation factors, the authors should indicate if the residues of EF-Tu contacting the SidH are conserved in the eukaryotic proteins.

While SidH is cytotoxic and apparently prevented extraction of SidH from human cells, the SidH(Δ -tail) is much less cytotoxic and could allow extraction of the bacterial protein with bound elongation factor from eukaryotic cells. Also, possibly using SidH_{Paris} N-term could be successful in extracting the complex.

Finally, the authors show binding of LubX in the context of the presence of EF-Tu and tRNA. Does LubX ubiquitinate SidH in the absence of EF-Tu? This could likely be tested with the mutants that do not bind EF-Tu and/or tRNA, which are already at hand. In other words, does EF-Tu binding involves any conformational change in SidH?

Can the ubiquitination site be narrowed down to a particular lysine within the Hb2 bundle? There are likely not many lysines exposed to the solvent in this segment. This could easily be verified experimentally.

Minor comments.

It would help to use three-letter codes for amino acids to avoid confusion with naming the tRNA bases.

The PDB identification codes for the two structures should be provided in the experimental section

and in the Table S1.

In the UniProt entry on SidH- Ipg2829 there is a reference to the MODBase model for SidH based on FANCI-FANCD2 complex (3S4W), also fully helical protein. Just, curious if there is any similarity between your structure and the model or the 3S4W PDB entry.

Reviewer #2 (Remarks to the Author):

This very interesting manuscript describes the characterization of the *Legionella pneumophila* T4SS substrate SidH. Previously it was shown that SidH is toxic when overexpressed in yeast cells and can be targeted for degradation within host cells by the metaeffector LubX. Since SidH has no sequence similarity to proteins of known function, the authors purified the protein and performed single particle analysis, thereby generating a 2.7Å structure containing a novel alpha helical arrangement. Interestingly, when SidH was purified from *E. coli*, it was found to be bound to the protein EF-Tu and a variety of t-RNAs. The authors proceeded to generate sidH mutations in residues at both the interfaces between SidH-tRNA and SidH-EF-Tu. These mutations abolished toxicity when SidH was overexpressed SidH human cells and in *Acanthamoeba castellanii*, suggesting the interaction was biologically relevant. The authors also generated a cryo-EM structure of SidH with LubX. Overall this is a very well-developed manuscript providing multiple interesting insights, specifically that SidH binds to EF-TU and tRNA.

Concerns:

1. The authors struggled to establish that SidH binds to EEF1A1, the human homolog of EF-TU, or to any tRNAs when SidH is expressed in host cells. They stated "But despite our efforts, we did not succeed in purifying SidH from HEK cells in sufficient purity and quantities for RNA extraction and sequencing. This is likely due to the high toxicity/cell death that SidH expression caused in HEK293T cells". This makes sense but it would really strengthen the paper if the authors could provide data demonstrating that SidH binds EEF1A1/tRNAs inside host cells. For example, have the authors tried to do an anti-SidH IP followed by an anti-EEF1A1 western using HEK cells lysates expressing SidH?

2. The authors state that sidH mutations that prevent interactions with EF-TU or tRNAs are no longer toxic when expressed in human cells and amoebae. However, unless I missed it, I don't recall seeing any data showing these mutant proteins are expressed and stable inside host cells. If not in the manuscript, this should be included.

3. Growth curves:

3a. Figure S18. Appears to be mis-graphed, i.e. look the 0 time point. Line starts between 10E-1 and 10E-2?

3b. Figure 7A. No evidence provided that the construct over-expresses wild-type SidH N-term.

3c. Figure 7A and 7B. Statistics need to be included, e.g. p values.

3d. Figure 7B. Control showing Δlpp2886 + vector should be included in this figure.

4. It is unclear how SidH is exported from Lp cells if it is bound to EF-TU/tRNAs. The authors should test if the interaction between SidH and EF-TU/tRNAs occurs when SidH is expressed at normal levels in Lp cells and/or provide an explanation for how secretion occurs in their Discussion.

5. The authors were unable to demonstrate an effect of SidH on protein synthesis, thus precluding determining why SidH binds EEF1A1/tRNAs. It would really benefit the manuscript if the authors could find a function for SidH, although this is likely outside the scope of this submission.

Reviewer #3 (Remarks to the Author):

This work by Sharma determined the structure of SidH, the largest known substrate of the *Legionella pneumophila* Dot/Icm system. They also obtained the structure of the SidH-LubX complex, thus providing a molecular mechanism of the ubiquitin E3 ligase LubX in substrate recognition. The entry of structural biologists into the field of bacterial effector biology has provided some tools highly complementary to the genetic and biochemical methods used by microbiologists to study microbial pathogenesis, and has offered many exciting insights into the mechanism of action of the effectors. Unfortunately, although the structures presented in this study is exciting and the authors should be commended for these feats, the lack of a definitive biochemical activity of SidH made it premature and less impactful.

Specific comments:

1. Protein translocation by T4SS such as the Dot/Icm system of *L. pneumophila* is believed to occur in which the protein substrate assumes a linear conformation in the transferring process (the size of the engaging pore and the translocon cannot accommodate fully folded proteins, particularly for large proteins like SidH). This raises the question whether the observed binding to bacterial EF-Tu (and tRNA) by SidH in protein purified from *E. coli* is physiologically meaningful. Does the interaction between EF-Tu (and the bacterial tRNA) and SidH occur in the cytosol of infected cells? if so, how can EF-Tu and tRNA reach the cytosol of infected cells? the authors should examine whether SidH interacts with eEF1A, the eukaryotic homolog of EF-Tu, and if so what is the impact of such interactions and whether SidH imposes any posttranslational modification on eEF-1A.
2. Lines 78-94 The discussion of the role of SdhA (thus inferring the potentially similar role for SidH?) in maintaining the integrity of infected cells. It is worth noting that these two proteins only share highly limited homology in a short region, and they may be functionally unrelated.
3. In *Lp* strain Paris, the *sidH* gene appeared to be split into two genes, *lpp2886* and *lpp2883*. Do these two proteins form a complex that functions similarly to the long single protein one found in the Philadelphia 1 strain?
4. Lines 74-76 the original study that identified the SidH family should be cited (PMID: 14715899).
5. Page 11, use Δ sidH instead of Δ lpp2886?

Reviewer #4 (Remarks to the Author):

I am very pleased to review such a very nice manuscript. It is very well written, structured and organized.

In summary

LP secretes effectors for intracellular replication. SidH, a toxic *LP* effector, was studied using cryo-EM. SidH showed a unique alpha helical structure and bound to an EF-Tu/t-RNA/GTP complex. Mutations disrupting SidH's interactions abolished its toxicity in human cells and *Acanthamoeba castellanii*. The cryo-EM structure of SidH with LubX, a regulator, provided insights into SidH's regulation. SidH mutants resistant to LubX-mediated degradation were more toxic in infection assays. This research enhances understanding of SidH's toxicity and its regulation during *LP* infection.

This manuscript focused intensively on SidH in all sides.

I like Fig. 2 and Fig. 4 and data behind these figures.
Fig 7 has very nice results and you can build up future plan on this idea.

Really, I have no comments on your work. Great work. Thank you

Reviewer #1 (Remarks to the Author):

The authors set to investigate the cellular function of the Legionella effector protein SidH, a paralog of an essential virulence factor SdhA, with which it shares ~40% sequence similarity over the N-terminal third of the protein. SidH is a 2225 amino acid long protein in *L. pneumophila* strain Philadelphia but in *L. pneumophila* strain Paris (and some other Legionella) strains the homolog of SidH is encoded by two genes, lpp2883 and lpp2886. At the later stage of infection SidH is a target for another Legionella effector, LubX, which ubiquitinates SidH, leading to its degradation.

Recombinant SidH that was expressed in *E. coli* purified with a tightly bound two other macromolecules. They were identified by sequencing as an *E. coli* elongation factor EF-Tu and a single molecule of tRNA. Subsequently, the authors determined the structure of the SidH (1-1620)–EF-Tu–tRNA by cryo-EM at the nominal resolution of 2.7 Å. The structure is made of eight α -helical bundles with sequence and structural similarities between N- and C-terminal segments, suggesting a gene duplication event during evolution. The SidH is toxic to human cells and this effect is predominantly associated with the C-terminal unmodeled segment. The binding sites of EF-Tu and tRNA on SidH were determined and further experiments with the homolog from the Paris strain showed that the formation of the EF-Tu–tRNA complex is a prerequisite for binding to SidH. Moreover, tRNA interacts with SidH predominantly through its T-loop which is highly conserved across all tRNAs and explains why many different tRNAs sequences were found by sequencing of the purified complex.

Based on the association of SidH with the *E. coli*'s EF-Tu and tRNA, the authors hypothesise that during infection SidH binds to the human elongation factor EEF1A1 or mitochondrial TUFM, and tRNA. Unfortunately, the attempts to purify SidH from human cells were unsuccessful, likely due to high toxicity (authors suggestion). To confirm the importance of EF-Tu and tRNA binding to SidH for its toxicity, the authors showed that mutating the residues crucial for their binding, based on the structure, leads to a diminished toxicity. The potential of SidH affecting protein synthesis through its interaction with the elongation factor 1 alpha-1 was considered, however. additional experiments showed no such effect of protein synthesis.

To investigate the impact of LubX on SidH the authors determined the cryo-EM structure of this complex. LubX binds to SidH in the presence of bound EF-Tu and tRNA, at a site distant from the latter binding sites. Their modeling of UBE2D2 onto LubX based the crystal structure of their complex, suggested that the ubiquitination site is located on the helical bundle Hb2. Mutating SidH residues participating in LubX binding led, indeed, to prolonged toxicity of this mutant when co-expressed with LubX as compared to the wt SidH.

Overall, while the authors have not identified the specific mechanism of SidH toxicity, their structural findings identified association with EF-Tu and tRNA, showed that the binding of EF-Tu and tRNA is essential for toxicity, showed that LubX binds at a site distant from EF-Tu and tRNA and indicated the Hb2 region as the most likely site for ubiquitination. These are important discoveries of general interest.

I have several comments.

1. Regarding the potential interactions of SidH with eukaryotic elongation factors, the authors should indicate if the residues of EF-Tu contacting the SidH are conserved in the eukaryotic proteins.

We thank the reviewer for this comment, we now include Figure S19 and S20 showing the sequence alignment of EF-Tu and human EEF1A1 and TUFM highlighting residue conservation. We also discuss this in the text (lines 509-512).

2. While SidH is cytotoxic and apparently prevented extraction of SidH from human cells, the SidH(delta-tail) is much less cytotoxic and could allow extraction of the bacterial protein with bound elongation

factor from eukaryotic cells. Also, possibly using SidHParis N-term could be successful in extracting the complex.

We thank the reviewer for this interesting suggestion. We thought about this experiment, but since SidHParis N-term loses all the toxicity of the FL SidH protein in human cells (Figure 4C), we believe that studying this fragment in isolation will not bring physiologically relevant insights into the function of SidH. Towards this end, we have ectopically expressed GFP-tagged FL SidH in HEK293T cells and instead of attempting to purify the protein for RNA extraction, we immunoprecipitated the GFP-SidH protein and analysed the co-eluted protein fraction using quantitative mass spectrometry (Figure 5A, text 325-333). Here, we identified almost all ribosomal proteins, several t-RNA ligases and many other RNA-binding proteins co-eluting with SidH. But we have not detected any human elongation factors in SidH enriched fraction.

We now discuss this in detail (lines 513-521) and state clearly that we have not managed to detect any binding between SidH and human tRNA and elongation factors directly in the host cell cytoplasm. We still like to remind that SidH binds to human tRNA in vitro (Figure 4D) and mutants deficient in binding to E.coli tRNA and EF-Tu lose toxicity in HEK293T cells and also during infection in Amoeba (Figure 5 and Figure 7). So there is a physiological relevance for the tRNA/EF-Tu binding region in SidH.

In light of these data, we are merely able to suggest that SidH is involved in some RNA regulatory pathway in cells but not propose a clear mechanism for the function of SidH. Accordingly, we are also toning down the title of our manuscript and we now call it “Structural basis for the toxicity of Legionella pneumophila effector SidH”. Precise role of SidH during infection, especially the identity of RNA that it binds and the process that it affects will be of active future investigations both by us and other labs. But we believe these are out of the scope of the current manuscript which reveals unique structures of SidH, SidH/LubX and offers a peek into the potential function of SidH through the bound E.coli tRNA and EF-Tu.

3. Finally, the authors show binding of LubX in the context of the presence of EF-Tu and tRNA. Does LubX ubiquitinate SidH in the absence of EF-Tu? This could likely be tested with the mutants that do not bind EF-Tu and/or tRNA, which are already at hand. In other words, does EF-Tu binding involves any conformational change in SidH?

We thank the reviewer for this comment. According to the reviewer’s suggestion, we performed in vitro ubiquitination assays with LubX and WT SidH or SidH HM (deficient in binding to tRNA and EF-TU) (Figure S15B). We found that both the WT and mutant are ubiquitinated by SidH. We went further ahead and identified sites of ubiquitination and the ubiquitin chain type assembled on SidH (Figure S17A, Table S3). We in fact also determined a low-resolution (6.8 Å) cryo-EM structure of SidH HM and found that there are no major structural changes in the molecule compared to SidH WT. Please see the figure below. Since the resolution of the map is not high enough and it does not offer any significant new insights, we did not include this low resolution map in the manuscript.

Figure: 6.8Å cryo-EM structure of SidH HM which cannot bind to tRNA and EF-Tu. The WT SidH model is morphed into the map.

4. Can the ubiquitination site be narrowed down to a particular lysine within the Hb2 bundle? There are likely not many lysines exposed to the solvent in this segment. This could easily be verified experimentally.

We have carried out an in vitro ubiquitination assay in the presence of LubX and SidH WT and then subjected it to LC-MS/MS to identify potential ubiquitination sites. The analysis revealed that lysine residues located at position K230 in Hb2; K358, K369 in the connecting loop between Hb2 and Hb3 and K656 in Hb4 bundle of SidH are the sites of ubiquitination (Figure S17A, Table S3). In agreement with our complex structure of SidH and LubX, the identified ubiquitination sites are close to the modeled E2 enzyme.

Minor comments.

1. It would help to use three-letter codes for amino acids to avoid confusion with naming the tRNA bases. *We thank the reviewer for this suggestion. We have now replaced the one letter code of amino acids (in the result section- "SidH/tRNA interaction") with three-letter codes to mitigate any potential confusion with the naming of tRNA bases.*

2. The PDB identification codes for the two structures should be provided in the experimental section and in the Table S1.

We are currently depositing our structures. Because of the uncertain tRNA (due to averaging of multiple tRNAs) identity in our model, the PDB staff is navigating us through this, but it is something that they also have not dealt before. We are committed to making the coordinates available for everyone before our paper is online. We are happy to share our pdb files and the map files with the reviewer meanwhile if required.

3. In the UniProt entry on SidH- Ipg2829 there is a reference to the MODBase model for SidH based on FANCI-FANCD2 complex (3S4W), also fully helical protein. Just, curious if there is any similarity between your structure and the model or the 3S4W PDB entry.

This MODBase model for the c-terminal segment of SidH (1108-2193aa) was constructed using PDB 3s4wA structure as a reference, albeit with only a 13% identity match. We compared a part of our SidH structure (1241-1619aa; Hb7 and Hb8) with the MODBase model, as the remainder of the SidH structure could not be built due to the local low-resolution of the map. Within this compared region, we noted a resemblance in terms of alpha helical nature, which the Reviewer had also highlighted. However, apart from this shared feature, we didn't observe other similarities, as there was no overlap between the two structures.

Reviewer #2 (Remarks to the Author):

This very interesting manuscript describes the characterization of the Legionella pneumophila T4SS substrate SidH. Previously it was shown that SidH is toxic when overexpressed in yeast cells and can be targeted for degradation within host cells by the metaeffector LubX. Since SidH has no sequence similarity to proteins of known function, the authors purified the protein and performed single particle analysis, thereby generating a 2.7Å structure containing a novel alpha helical arrangement. Interestingly, when SidH was purified from E. coli, it was found to be bound to the protein EF-Tu and a variety of t-RNAs. The authors proceeded to generate sidH mutations in residues at both the interfaces between SidH-tRNA and SidH-EF-Tu. These mutations abolished toxicity when SidH was overexpressed SidH human cells and in Acanthamoeba castellanii, suggesting the interaction was biologically relevant. The authors also generated a cryo-EM structure of SidH with LubX. Overall this is a very well-developed manuscript providing multiple interesting insights, specifically that SidH binds to EF-TU and tRNA.

Concerns:

1. The authors struggled to establish that SidH binds to EEF1A1, the human homolog of EF-TU, or to any tRNAs when SidH is expressed in host cells. They stated “But despite our efforts, we did not succeed in purifying SidH from HEK cells in sufficient purity and quantities for RNA extraction and sequencing. This is likely due to the high toxicity/cell death that SidH expression caused in HEK293T cells”. This makes sense but it would really strengthen the paper if the authors could provide data demonstrating that SidH binds EEF1A1/tRNAs inside host cells. For example, have the authors tried to do an anti-SidH IP followed by an anti-EEF1A1 western using HEK cells lysates expressing SidH?

We thank the reviewer for this comment. Towards this end, we have ectopically expressed GFP-tagged FL SidH in HEK293T cells and instead of attempting to purify the protein for RNA extraction, we immunoprecipitated the GFP-SidH protein and analysed the co-eluted protein fraction using quantitative mass spectrometry (Figure 5A, text 325-333). Here, we identified almost all ribosomal proteins, several t-RNA ligases and many other RNA-binding proteins co-eluting with SidH. But we have not detected any human elongation factors in SidH enriched fraction.

We now discuss this in detail (lines 513-521) and state clearly that we have not managed to detect any binding between SidH and human tRNA and elongation factors directly in the host cell cytoplasm. We still like to remind that SidH binds to human tRNA in vitro (Figure 4D) and mutants deficient in binding to E.coli tRNA and EF-Tu lose toxicity in HEK293T cells and also during infection in Amoeba (Figure 5 and Figure 7). So there is a physiological relevance for the tRNA/EF-Tu binding region in SidH.

In light of these data, we are merely able to suggest that SidH is involved in some RNA regulatory pathway in cells but not propose a clear mechanism for the function of SidH. Accordingly, we are also toning down the title of our manuscript and we now call it “Structural basis for the toxicity of Legionella pneumophila effector SidH”. Precise role of SidH during infection, especially the identity of RNA that it binds and the process that it affects will be of active future investigations both by us and other labs. But we believe these are out of the scope of the current manuscript which reveals unique structures of SidH, SidH/LubX and offers a peek into the potential function of SidH through the bound E.coli tRNA and EF-Tu.

2. The authors state that sidH mutations that prevent interactions with EF-TU or tRNAs are no longer toxic when expressed in human cells and amoebae. However, unless I missed it, I don't recall seeing any data showing these mutant proteins are expressed and stable inside host cells. If not in the manuscript, this should be included.

We thank the reviewer for pointing out this crucial missing data. We have now included WBs showing expression of WT and mutant proteins in HEK cells (Figure S12C) and also in Legionella pneumophila Paris strains used for infection experiments (Figure S18B).

3. Growth curves:

3a. Figure S18. Appears to be mis-graphed, i.e. look the 0-time point. Line starts between $10E-1$ and $10E-2$?

The graph initiates with a value of 10^0 at the 0-time point. However, there is a subsequent decline in the bacterial count at the 1-hour time point, causing the value on the y-axis to drop to the range between 10^{-1} and 10^{-2} . Since this time range here is condensed tightly on the X-axis it appears as though it is mis-graphed, although it is not.

3b. Figure 7A. No evidence provided that the construct over-expresses wild-type SidH N-term.

We thank the reviewer for bringing this to our attention. In response to your comment, we have included the western blot data showing the over expression of SidH N-term WT and mutants (Figure S18B).

3c. Figure 7A and 7B. Statistics need to be included, e.g. p values.

We have now added the p values to both the graphs in the figure itself and mentioned the statistical analysis performed in the figure legend.

3d. Figure 7B. Control showing Δ lpp2886 + vector should be included in this figure.

Due to the complexity of this CFU experiment involving biological triplicates for each condition and number of time points, it was practically only possible to do infection with 4 strains in one experiment and since we performed Δ lpp2886 + vector control in Figure 7A, we had to prioritize the SidH mutant complementation experiments in Figure 7B and omit the vector control. We hope the reviewer understands these practical difficulties and agrees that it would not change the conclusions drawn.

4. It is unclear how SidH is exported from Lp cells if it is bound to EF-TU/tRNAs. The authors should test if the interaction between SidH and EF-TU/tRNAs occurs when SidH is expressed at normal levels in Lp cells and/or provide an explanation for how secretion occurs in their Discussion.

We thank the reviewer for this insightful comment. We now include text in the discussion about the secretion of SidH (lines 501-509). We agree with the reviewer that EF-Tu and tRNAs from E.coli share high sequence similarity to the counterparts in LP, it is likely that these molecules are bound to SidH within LP. It is worth noting that a recent paper (PMID: 32457311) showed that the extreme C-terminus of SidH which is very distant from the t-RNA binding site serves as the secretion signal. This indicates that even in the “bound state” SidH will be able to interact with the components of the secretion system, undergo unfolding and eventually secretion.

5. The authors were unable to demonstrate an effect of SidH on protein synthesis, thus precluding determining why SidH binds EEF1A1/tRNAs. It would really benefit the manuscript if the authors could find a function for SidH, although this is likely outside the scope of this submission.

Indeed, we have made a lot of unsuccessful attempts over the last couple of years in delineating the physiological role of SidH and role of tRNA binding in it. With the most recent data in the form of quantitative mass spectrometry (Figure 5A), we could hint at the co-existence of SidH with ribosomal and other RNA-binding proteins including tRNA ligases. We will continue to work on elucidating the precise physiological role of SidH through Cross-Linking ImmunoPrecipitation Sequencing (CLIP seq) experiments using the infection lysates to unbiasedly identify the RNA molecules bound to SidH. We would also like to probe the effect of SidH on protein synthesis-unrelated functions of tRNA including in apoptosis (PMID: 20227371). But for the scope of the current manuscript, we hope the reviewer agrees that we are revealing unique insights into the structurally and functionally uncharacterized LP effector SidH which also has broad implications in SdhA family of effectors.

Reviewer #3 (Remarks to the Author):

This work by Sharma determined the structure of SidH, the largest known substrate of the Legionella pneumophila Dot/Icm system. They also obtained the structure of the SidH-LubX complex, thus providing a molecular mechanism of the ubiquitin E3 ligase LubX in substrate recognition. The entry of structural biologists into the field of bacterial effector biology has provided some tools highly complementary to the genetic and biochemical methods used by microbiologists to study microbial pathogenesis, and has offered many exciting insights into the mechanism of action of the effectors. Unfortunately, although the structures presented in this study is exciting and the authors should be

commended for these feats, the lack of a definitive biochemical activity of SidH made it premature and less impactful.

Specific comments:

1. Protein translocation by T4SS such as the Dot/Icm system of *L. pneumophila* is believed to occur in which the protein substrate assumes a linear conformation in the transferring process (the size of the engaging pore and the translocon cannot accommodate fully folded proteins, particularly for large proteins like SidH). This raises the following question-

- a. Whether the observed binding to bacterial EF-Tu (and tRNA) by SidH in protein purified from *E. coli* is physiologically meaningful.

We have shown that mutant versions of SidH deficient in binding to tRNA and/or EF-Tu cannot functionally complement the toxicity exerted by the WT SidH both in human cells and in amoeba during infection (Figure 5 and Figure 7). This underlines that the region of SidH involved in binding to these factors is physiologically relevant. However, we are currently unable to explain the precise role of this functionally important part of SidH. We have now included new quantitative proteomics data showing the interactome of GFP-SidH in human cells (Figure 5A). This revealed that immunoprecipitated SidH comes enriched with almost all the ribosomal proteins, several tRNA ligases and other RNA-binding proteins. From the structure of SidH bound to tRNA and EF-Tu, these associations of SidH in human cells make sense but it is important to point out here that we do not detect human elongation factors enriched in the SidH-bound fraction. We clearly point out this limitation of our study now in discussion (lines 513-521). Accordingly, we are also toning down the title of our manuscript and we now call it "Structural basis for the toxicity of Legionella pneumophila effector SidH".

*Precise role of SidH during infection, especially the identity of RNA that it binds and the process that it affects will be of active future investigations both by us and other labs. But we believe these are out of the scope of the current manuscript which reveals unique structures of SidH, SidH/LubX and offers a peek into the potential function of SidH through the bound *E. coli* tRNA and EF-Tu.*

- b. Does the interaction between EF-Tu (and the bacterial tRNA) and SidH occur in the cytosol of infected cells? if so, how can EF-Tu and tRNA reach the cytosol of infected cells?

*We thank the reviewer for this insightful comment. This is in principle plausible but we prefer the idea that SidH rather uses these binding properties which enabled its binding to *E. coli* tRNA/EF-Tu to interact with as yet unidentified RNA and protein factors in the host cytosol. We favour this notion because of the following two factors. Firstly, we observed SidH interacting with RNA associated factors in our quantitative mass spectrometry experiment (Figure 5A) and secondly SidH elicits toxicity in HEK cells that is dependent on tRNA-binding region (Figure 5B). For the full functional analysis of SidH, we believe, further experiments such as Cross-Linking ImmunoPrecipitation Sequencing (CLIP seq) have to be conducted to identify the RNA molecules bound to SidH in an unbiased manner. It would also be interesting to probe the effect of SidH on protein synthesis-unrelated functions of tRNA including in apoptosis (PMID: 20227371).*

Regarding the argument whether tRNA of LP reaches host cytosol, it has been shown recently by Sahr et al., that they are transported into the host cytosol through extracellular vesicles (PMID: 35140216). However bacterial EF-Tu has not shown to be translocated to the host cytosol at least in the case of LP but has been shown to do so in the case of plant pathogens (PMID: 15548740).

- c. The authors should examine whether SidH interacts with eEF1A, the eukaryotic homolog of EF-Tu, and if so what is the impact of such interactions and whether SidH imposes any posttranslational modification on eEF-1A.

We thank the reviewer for this comment. In our new interaction proteomics data (Figure 5A) we did not detect eEF1A as a factor enriched in IPed SidH fraction. We have included discussion on this data in light of the potential physiological role of SidH (lines 513-521)

2. Lines 78-94 The discussion of the role of SdhA (thus inferring the potentially similar role for SidH?) in maintaining the integrity of infected cells. It is worth noting that these two proteins only share highly limited homology in a short region, and they may be functionally unrelated.

We thank the reviewer for this comment and point to a section in the introduction where we describe precisely this aspect (lines 78-83). “SdhA belongs to the SdhA family of effectors sharing ~40% sequence similarity with its paralogs SdhB and SidH in the N-terminal region (~700 amino acids) of the protein sequence. Interestingly, deletion of SdhA alone results in defective growth in mouse macrophages while deletion of the two paralogues SidH and SdhB does not cause any significant growth phenotype indicating that these paralogs may have different functions during infection.”

3. In Lp strain Paris, the sidH gene appeared to be split into two genes, lpp2886 and lpp2883. Do these two proteins form a complex that functions similarly to the long single protein one found in the Philadelphia 1 strain?

We thank the reviewer for this comment. Indeed, we tested this already. In Figure 4B, we show that these two fragments of Lp strain Paris SidH do interact and in Figure 4C, we show that only the co-transfection of these fragments results in toxicity comparable to that of FL SidH from LP Philadelphia, where as the expression of individual fragments in HEK cells do not elicit any toxicity.

4. Lines 74-76 the original study that identified the SidH family should be cited (PMID: 14715899).

We thank the reviewer for pointing this out, we have now added this reference.

5. Page 11, use Δ sidH instead of Δ lpp2886?

In the Paris strain of Legionella pneumophila, the SidH is split into two halves (lpp2886 and lpp2883). The knockout strain of LP Paris we used here lacked only the lpp2886 (n-terminal part) of SidH. Therefore, to be accurate, we referred to this strain as Δ lpp2886.

Reviewer #4 (Remarks to the Author):

I am very pleased to review such a very nice manuscript. It is very well written, structured and organized.

In summary

LP secretes effectors for intracellular replication. SidH, a toxic LP effector, was studied using cryo-EM. SidH showed a unique alpha helical structure and bound to an EF-Tu/t-RNA/GTP complex. Mutations disrupting SidH's interactions abolished its toxicity in human cells and Acanthamoeba castellanii. The cryo-EM structure of SidH with LubX, a regulator, provided insights into SidH's regulation. SidH mutants resistant to LubX-mediated degradation were more toxic in infection assays. This research enhances understanding of SidH's toxicity and its regulation during LP infection.

This manuscript focused intensively on SidH in all sides.

I like Fig. 2 and Fig. 4 and data behind these figures.

Fig 7 has very nice results and you can build up future plan on this idea.

Really, I have no comments on your work. Great work. Thank you

We sincerely thank the reviewer for the kind words. We hope the revised version further improves the accuracy and conclusions of the manuscript.

REVIEWERS' COMMENTS

Reviewer #1 (Remarks to the Author):

The authors addressed most of my comments. One remaining exception is to use a three-letter code for the amino acids throughout the text. This is not the case on page 11.

Reviewer #2 (Remarks to the Author):

I am satisfied with the revisions and believe the manuscript is ready for publication.

Reviewer #3 (Remarks to the Author):

The authors have adequately addressed my concerns, I am sure that some of the remaining questions will be answered in future studies by this group or others based on the solid foundation formed by this exciting structural study.

Point by Point Response

Reviewer #1 (Remarks to the Author):

The authors addressed most of my comments. One remaining exception is to use a three-letter code for the amino acids throughout the text. This is not the case on page 11.

We have made all the necessary changes asked by the reviewer. we thank the reviewer for the valuable feedback and for taking the time to review our manuscript. We are pleased to hear that the reviewer found our work satisfactory.

Reviewer #2 (Remarks to the Author):

I am satisfied with the revisions and believe the manuscript is ready for publication.

We thank the reviewer for the valuable feedback and for taking the time to review our manuscript. We are pleased to hear that the reviewer found our work satisfactory.

Reviewer #3 (Remarks to the Author):

The authors have adequately addressed my concerns, I am sure that some of the remaining questions will be answered in future studies by this group or others based on the solid foundation formed by this exciting structural study.

We thank the reviewer for the valuable feedback and for taking the time to review our manuscript. We are pleased to hear that the reviewer found our work satisfactory.